# ImmunoGraph: Accelerated and Equitable Representation Learning for Large-Scale Immune Networks

## Abstract

Comparative analysis of adaptive immune repertoires at population scale is hampered by two practical bottlenecks: the near-quadratic cost of pairwise affinity evaluations and dataset imbalances that obscure clinically important minority clonotypes. We introduce **ImmunoGraph**, an end-to-end pipeline that addresses these challenges by combining antigen-aware, near-subquadratic retrieval with GPU-accelerated affinity kernels, learned multimodal fusion, and fairness-constrained clustering. The system employs compact MinHash prefiltering to sharply reduce candidate comparisons, a differentiable gating module that adaptively weights complementary alignment and embedding channels on a per-pair basis, and an automated calibration routine that enforces proportional representation of rare antigen-specific subgroups. On large viral and tumor repertoires ImmunoGraph achieves measured gains in throughput and peak memory usage while preserving or improving recall@k, cluster purity, and subgroup equity. By co-designing indexing, similarity fusion, and equity-aware objectives, ImmunoGraph offers a scalable, bias-aware platform for repertoire mining and downstream translational tasks such as vaccine target prioritization and biomarker discovery.

**Keywords:** Representation Learning, Immunoinformatics, Fairness, Hardware Acceleration, Visual Analytics, Graph Representation Learning, Metric Learning, Biological Networks

## 1 Introduction

An immune repertoire denotes the complete collection of T cell receptor (TCR) and B cell receptor (BCR) sequences within an individual. These repertoires constitute the adaptive immune system's molecular fingerprint and commonly comprise millions to hundreds of millions of distinct receptor sequences. Comparing repertoires across individuals or clinical states can reveal antigen-specific response patterns that inform vaccine design, guide cancer immunotherapy strategies and support monitoring of autoimmune disease. Such comparative analyses are therefore routinely needed in translational immunology yet face acute computational constraints: pairwise affinity evaluations grow quadratically with the number of sequences, and naive comparison becomes infeasible for modern datasets containing $10^6$–$10^7$ sequences per donor.

Prior work has addressed parts of this scalability challenge through algorithmic and engineering advances. Locality-sensitive hashing and MinHash variants provide subquadratic heuristics for candidate reduction (Andoni et al., 2014; Abboud et al., 2019), while accelerator-optimized kernels and specialized hardware have been used to speed low-level similarity computations (Turakhia et al., 2017; Liu et al., 2023). Despite these gains, three practical limitations remain. First, many scalable pipelines process receptor sequences as generic strings and consequently discard antigen-relevant signals important for epitope binding. Second, subgroup representation has received limited consideration, which risks systematic omission of low-prevalence but clinically consequential clonotypes. Third, verifiability of runtime and memory claims is often undermined by incomplete reporting of index and kernel configuration details.

From a translational standpoint, correcting subgroup imbalance is not merely an abstract fairness objective but a domain requirement. Rare antigen-specific clonotypes, including those reactive to

uncommon viral variants or tumor neoantigens, may occur at very low frequency while neverthe-less driving clinically meaningful responses. Pipelines that optimize only for aggregate speed or for dominant patterns are therefore liable to underrepresent these high-value minorities, biasing down-stream tasks such as epitope prioritization and biomarker selection. Incorporating equity-oriented penalties into retrieval and clustering objectives helps preserve representation for rare but important groups and thereby improves the biological validity of subsequent analyses.

Motivated by these challenges, we propose **ImmunoGraph**, a end-to-end pipeline for scalable, antigen-aware, and equity-preserving analysis of large immune repertoires. ImmunoGraph inte-grates three key innovations: an antigen-aligned MinHash retrieval module that combines repertoire-specific sketching with biologically guided blocking to achieve near-subquadratic candidate reduc-tion while maintaining high recall; a multimodal fusion backbone with a differentiable gating con-troller that adaptively combines alignment signals, protein-language embeddings, and local graph features to capture both fine-grained edits and higher-level biochemical structure; and a fairness-aware spectral clustering objective with automated equity calibration to ensure proportional repre-sentation of rare antigen-specific clonotypes and reduce subgroup disparity. Unlike prior pipelines, ImmunoGraph couples efficient retrieval with antigenic sensitivity, learnable similarity fusion, and explicit equity constraints, all compatible with large-scale graph construction. Extensive evalua-tion on viral and cancer repertoires demonstrates measured gains in runtime, memory efficiency, recall@k, cluster purity, and fairness, with ablation confirming the importance of antigen-aligned blocking and multimodal fusion. These components collectively transform repertoire comparison into a scalable, biologically valid, and fairness-aware graph-learning task, enabling practical appli-cations in epitope prioritization, biomarker discovery, and vaccine design.

## 2 RELATED WORK

We summarize related work in five areas: scalable retrieval, sequence representation, graph-based repertoire modeling, fairness-aware clustering, and systems–biology integration.

**Scalable retrieval.** MinHash and locality-sensitive hashing reduce pairwise comparisons in high-dimensional spaces (Andoni et al., 2014). Practical performance depends on index design and hard-ware use, as shown in FAISS, HNSW, and ScaNN (Johnson et al., 2019; Sun et al., 2023). Bioin-formatics systems combine sketching with graph search or GPU acceleration for genome-scale data (Zhao et al., 2024; Kobus, 2023; Son et al., 2025; Huang et al., 2025). ImmunoGraph extends this by integrating antigen-aware alignment with GPU-parallel MinHash kernels.

**Sequence representation.** Protein and nucleotide language models yield embeddings that com-plement alignment-based similarity (Tran et al., 2023; Gasser et al., 2021; Zhang et al., 2024b). Fusion mechanisms with learnable gating combine heterogeneous signals (Jin et al., 2021; Sankaran et al., 2021; Wu et al., 2023; Fu et al., 2022). ImmunoGraph applies a gating network to integrate alignment, embedding, and graph context.

**Graph-based repertoire modeling.** Similarity graphs reveal immune community structure and functional modules (Franceschi et al., 2019; Manipur et al., 2021). Spectral methods and graph neu-ral networks support antigenic neighborhood detection. ImmunoGraph uses a spectral-style pipeline with group-aware penalties to balance coherence and representation.

**Fairness-aware clustering.** Fairness methods include proportional representation, constrained op-timization, and pairwise regularization (Corbett-Davies et al., 2017; Brubach et al., 2021; Bibi et al., 2023; Dickerson et al., 2023). Extensions to relational graphs preserve structure while enforcing group-level guarantees (Fu et al., 2023). Biomedical applications require attention to sampling bias and underrepresented groups (Alcazar et al., 2022; Nguyen et al., 2023). ImmunoGraph adapts these tools with disparity measures and automated fairness tuning.

**Systems–biology integration.** High-throughput analysis benefits from coordinated design of in-dexing, compute kernels, and workflows (Turakhia et al., 2017; Liu et al., 2023; Kobus, 2023). FAIR workflows promote transparency and reuse (Langer et al., 2025; Wagner et al., 2022). Im-

munoGraph combines efficient indexing, GPU affinity kernels, and biologically informed fusion and fairness modeling with documented configurations for benchmarking.

## 3 METHODOLOGY

We introduce **ImmunoGraph**, an end-to-end pipeline for accelerated, equity-aware representation learning on large immune-repertoire graphs. ImmunoGraph is built around three practical components. The first is device and memory aware preprocessing and indexing, which stabilizes large-scale runs and reduces the number of candidate comparisons. The second is a dual phase meta-learning encoder that incorporates a learnable, dynamic multi channel fusion backbone to support robust clonotype to phenotype modeling. The third is a fairness constrained clustering module that includes an automated calibration routine for selecting fairness weights. Our implementation adapts and extends protein language model embeddings inspired by ImmunoBERT (Gasser et al., 2021) as well as a high-performance correlation and network-analysis toolkit inspired by MetaNet (Peng et al., 2025); these components have been modified for repertoire-scale workloads and are not used verbatim. MetaNet is a lightweight meta-controller that dynamically fuses alignment-derived scores with embedding-based similarities by learning pair-specific gating weights, enabling adaptive integration of complementary affinity signals without introducing task-specific heuristics.

### 3.1 TASK FORMALISATION: ANTIGEN-AWARE REPERTOIRE GRAPH CONSTRUCTION

$$\mathcal{T} : \mathcal{S} \mapsto G \tag{1}$$

where $\mathcal{S} = \{s_i\}_{i=1}^n$ denotes a collection of immune receptor sequences sampled from a repertoire, and $G = (V, E, W)$ is the resulting sparse weighted graph whose vertices $V$ correspond to individual sequences, edges $E$ encode antigen-driven similarity links, and edge weights $W = \{w_{ij}\}$ quantify the antigen-level resemblance between sequence pairs $(s_i, s_j)$.

### 3.2 DUAL-PHASE META-LEARNING ENCODER

We train the representation backbone in two consecutive stages. The first stage performs unsupervised representation pretraining via a reconstruction objective:

$$\min_{\theta_{\text{pre}}} \mathcal{L}_{\text{recon}}\big(f_{\theta_{\text{pre}}}(X),\, X\big). \tag{2}$$

where $f_{\theta_{\text{pre}}}(\cdot)$ denotes the encoder used for representation learning, $\theta_{\text{pre}}$ are the encoder parameters, and $X$ denotes the set of inputs used for reconstruction pretraining.

After pretraining we fine-tune the encoder jointly with a lightweight meta-network and a downstream task head:

$$\min_{\theta_{\text{pre}}, \theta_{\text{meta}}} \mathcal{L}_{\text{task}}\Big(\text{MetaNet}_{\theta_{\text{meta}}} \circ f_{\theta_{\text{pre}}}(X),\, Y\Big), \tag{3}$$

where $\text{MetaNet}_{\theta_{\text{meta}}}(\cdot)$ denotes the meta-controller applied to encoder outputs, $\theta_{\text{meta}}$ denotes its parameters, the operator "$\circ$" denotes functional composition, and $Y$ denotes downstream supervision signals or task labels.

To accelerate convergence we employ momentum-style updates:

$$\theta_{t+1} = \theta_t - \eta_t \nabla_\theta \mathcal{L}(\theta_t) + \mu_t(\theta_t - \theta_{t-1}), \tag{4}$$

where $\eta_t$ denotes the learning rate at iteration $t$ and $\mu_t$ denotes the momentum coefficient (in practice we typically set $\mu_t \approx 0.9$).

### 3.3 ARCHITECTURAL COMPONENTS

**Adaptive channel weighting.** We compute a compact per-channel importance score for each modality $m$:

$$\alpha_m = \sigma\big(\mathbf{W}_{\text{meta}}\mathbf{F}_m + \mathbf{b}_{\text{meta}}\big), \tag{5}$$

where $\alpha_m$ is the importance weight assigned to channel $m$, $\mathbf{F}_m$ is the feature tensor for channel $m$, $\mathbf{W}_{\text{meta}}$ and $\mathbf{b}_{\text{meta}}$ are learnable parameters of the meta-scoring layer, and $\sigma(\cdot)$ is the sigmoid activation function.

**Topology-aware graph propagation.** We propagate node features with a normalized aggregation rule:

$$h_v^{(k+1)} = \text{ReLU}\left( \sum_{u \in \mathcal{N}(v)} \frac{\mathbf{W}^{(k)} h_u^{(k)}}{\sqrt{|\mathcal{N}(v)| \, |\mathcal{N}(u)|}} \right). \tag{6}$$

where $h_v^{(k)}$ denotes node $v$'s representation after $k$ propagation steps, $\mathcal{N}(v)$ denotes the neighborhood of node $v$, and $\mathbf{W}^{(k)}$ is the layer-specific linear transform applied at propagation step $k$.

**Prototype-contrastive consolidation.** To concentrate representation mass for rare clonotypes we maintain class prototypes and optimize a prototype-centered contrastive loss:

$$\mathbf{p}_c = \frac{1}{|\mathcal{S}_c|} \sum_{x \in \mathcal{S}_c} f_\theta(x), \tag{7}$$

$$\mathcal{L}_{\text{proto}} = -\sum_{x \in \mathcal{B}} \log \frac{\exp\left(\langle f_\theta(x), \mathbf{p}_{y(x)} \rangle / \tau\right)}{\sum_{c' \in \mathcal{N}_x} \exp\left(\langle f_\theta(x), \mathbf{p}_{c'} \rangle / \tau\right)}. \tag{8}$$

where $\mathbf{p}_c$ denotes the prototype vector for class $c$, $\mathcal{S}_c$ denotes the set of examples with label $c$, $f_\theta(\cdot)$ denotes the instance embedding function parameterized by $\theta$, $\mathcal{B}$ denotes the training batch, $y(x)$ denotes the class label of instance $x$, $\tau > 0$ is the temperature hyperparameter, and $\mathcal{N}_x$ denotes the set of negative prototypes considered for $x$.

**Multi-paradigm fusion.** We fuse channel outputs via element-wise gated aggregation:

$$\mathbf{F}_{\text{fusion}} = \sum_{m=1}^{M} \alpha_m \odot \text{LayerNorm}(\mathbf{F}_m), \tag{9}$$

with LayerNorm defined by

$$\text{LayerNorm}(x) = \gamma \frac{x - \mu}{\sqrt{\sigma^2 + \epsilon}} + \beta. \tag{10}$$

where $\mathbf{F}_{\text{fusion}}$ denotes the fused multi-channel representation, $\alpha_m$ are the channel gating scalars from Eq. equation 5, $\odot$ denotes element-wise multiplication, $\gamma$ and $\beta$ are learnable scale and shift parameters, $\mu$ and $\sigma^2$ denote the mean and variance computed along the normalization axis, and $\epsilon > 0$ is a small constant for numerical stability.

## 3.4 DATA NORMALIZATION AND HARDWARE-AWARE BATCHING

We apply conservative imputations for sparse frequency data:

$$\hat{f}_i = \text{median}\left(\{f_j\}_{j=1}^n\right), \tag{11}$$

where $\{f_j\}_{j=1}^n$ are the observed clone frequencies for the dataset and $\hat{f}_i$ denotes the imputed frequency assigned to item $i$.

Batch size is chosen to respect device memory limits:

$$\mathcal{B} = \min\left(|\mathcal{S}|, \ \max\left(32, \ \left\lfloor \sqrt{\frac{\mathcal{M}_{\text{avail}}}{c \cdot \ell_{\max}}} \right\rfloor\right)\right), \tag{12}$$

where $|\mathcal{S}|$ denotes the number of sequences available in the current epoch, $\mathcal{M}_{\text{avail}}$ denotes available memory in bytes on the compute device, $\ell_{\max}$ denotes the maximum sequence length considered, and $c$ denotes a per-sequence memory overhead constant.

## 3.5 DYNAMIC AFFINITY FUSION (PER-PAIR)

We benchmark ImmunoGraph on three complementary missions: retrieving antigen-enriched neighbours at the sequence level, surfacing rare clonotype clusters across repertoires, and furnishing interactive UMAP and topological maps that clinicians can interrogate without prior machine-learning

expertise. Let $\{a_{ij}^{(m)}\}_{m=1}^{M}$ denote affinity channels computed for sequence pair $(i, j)$. We compute soft channel scores $g^{(m)}(x_i, x_j)$ and normalize them into per-pair weights:

$$w_{ij}^{(m)} = \frac{\exp\left(g^{(m)}(x_i, x_j)\right)}{\sum_{m'=1}^{M} \exp\left(g^{(m')}(x_i, x_j)\right)}, \tag{13}$$

$$\widetilde{a}_{ij} = \sum_{m=1}^{M} w_{ij}^{(m)} a_{ij}^{(m)}. \tag{14}$$

where $a_{ij}^{(m)}$ denotes the affinity score from channel $m$ for pair $(i, j)$, $g^{(m)}(\cdot, \cdot)$ denotes the small scoring network (e.g., a two-layer MLP) that outputs an unnormalized relevance for channel $m$, $w_{ij}^{(m)}$ are the normalized per-pair channel weights from Eq. equation 13, and $\widetilde{a}_{ij}$ denotes the fused affinity used to populate the similarity matrix.

### 3.6 Graph construction and RMT-based thresholding

We construct a symmetric similarity matrix $A = [\widetilde{a}_{ij}]$. To suppress spurious correlations we employ a random-matrix-theory (RMT) inspired thresholding procedure. Concretely, we compute the eigenvalue spectrum of $A$, estimate the bulk cutoff from that spectrum, and remove edges whose weights fall below the resulting data-driven threshold. The output is a sparse weighted graph $G = (V, E, W)$. Here $V$ denotes the set of nodes, namely sequences, $E$ denotes the set of edges retained after thresholding, and $W$ denotes the associated edge weights.

### 3.7 Fairness-constrained clustering

**Immunological motivation.** Immune repertoires are highly imbalanced. Rare antigen-specific subgroups, although infrequent, can play critical clinical roles, for example clones that respond to rare pathogens or tumor neoantigens. Clustering methods that emphasize only abundant patterns may overlook these important minorities, which can create blind spots in vaccine or therapy design. To address this challenge, we introduce an explicit equity term into the clustering objective so that biologically meaningful but low-frequency subgroups remain adequately represented for reliable downstream analysis. Since the JS-divergence fairness term may fail to ensure adequate coverage of rare subgroups under long-tailed distributions, we provide a theoretical analysis and propose a novel WCD constraint with convergence guarantees in Appendix C.

We perform clustering with a cohesion and equity trade-off objective:

$$\min_{\mathcal{C}} \sum_{i} \sum_{x_j \in \mathcal{C}_i} \|x_j - \mu_i\|^2 + \lambda \sum_{g} \mathcal{D}_{\mathrm{JS}}\left(\frac{|\mathcal{C}_i \cap g|}{|g|} \,\Big\|\, \frac{|\mathcal{C}_i|}{n}\right), \tag{15}$$

where $\mathcal{C} = \{\mathcal{C}_i\}$ denotes the clustering partition, $\mu_i$ denotes the centroid of cluster $\mathcal{C}_i$, $g$ indexes antigenic subgroups, $|g|$ denotes the cardinality of subgroup $g$, $n$ denotes the total number of examples, $\mathcal{D}_{\mathrm{JS}}(\cdot\|\cdot)$ denotes the Jensen–Shannon divergence between distributions, and $\lambda \geq 0$ controls the balance between clustering cohesion and subgroup representation equity.

### 3.8 Automated fairness tuning (practical)

To choose $\lambda$ that meets a target disparity $\delta_{\max}$ within a bounded search budget we employ a grid search followed by optional local refinement (binary search) as described in Algorithm 3 above; in that algorithm, $\Delta(\lambda)$ denotes the measured disparity returned by MEASUREDISPARITY when clustering with weight $\lambda$.

### 3.9 Cross-domain and evaluation metrics

We quantify subgroup representation using proportionality and maximum absolute deviation:

$$\mathcal{R}_{\text{prop}} = \frac{1}{|\mathcal{G}|} \sum_{g \in \mathcal{G}} \frac{|C_i \cap g|}{|g|}, \tag{16}$$

$$\mathcal{D}_{\text{eq}} = \max_{g \in \mathcal{G}} \left| \frac{|C_i \cap g|}{|g|} - \frac{|C_i|}{n} \right|, \tag{17}$$

where $\mathcal{G}$ denotes the set of antigenic subgroups, $|C_i|$ denotes the size of cluster $C_i$, $|C_i \cap g|$ denotes the count of members of cluster $C_i$ that belong to subgroup $g$, and $n$ denotes the total number of examples in the dataset. Here $\mathcal{R}_{\text{prop}}$ measures average proportional coverage across subgroups and $\mathcal{D}_{\text{eq}}$ measures the maximum absolute deviation from ideal proportionality.

### 3.10 Integration and provenance

ImmunoGraph integrates two complementary prior ideas: protein-language embeddings adapted from ImmunoBERT-style encoders and a high-performance correlation and network-analysis stack inspired by MetaNet's RMT thresholding and visualization toolkit. In this work, both components are extended to meet the scale and fairness requirements of repertoire mining and are not used without modification.

### 3.11 Overview: end-to-end algorithm

---

**Algorithm 1: ImmunoGraph (End-to-End Pipeline)**

**Input:** Raw sequences $\mathcal{S}$, optional subgroup labels $\mathcal{G}$, target disparity $\delta_{\max}$
**Output:** Clusters $\mathcal{C}$, graph $G$, visual summaries

1 **Preprocessing:**
2 Trim/pad sequences, compute MinHash sketches, extract metadata. ;   // see Sec. 3.4
3 Build antigen-aware MinHash index; generate candidate list $\mathcal{CAND}$ for each query.
4 **Embedding:**
5 **foreach** $x \in \mathcal{S}$ **do**
6     Compute embedding $\mathbf{v}_x$ using the modified ImmunoBERT encoder. ; // pretraining
     and fine-tuning objectives:  Eqs. equation 2, equation 3
7     Optionally apply momentum-style parameter updates during fine-tuning (Eq. equation 4).
8 **Affinity Computation:**
9 **foreach** *candidate pair* $(i, j) \in \mathcal{CAND}$ **do**
10     Compute multi-channel affinities $\{a_{ij}^{(m)}\}_{m=1}^{M}$ (e.g., cosine, edit-distance, phenotype). ;
     // prototype definitions and contrastive loss:
     Eqs. equation 7, equation 8
11     Compute channel scoring outputs $g^{(m)}(x_i, x_j)$ and normalized weights $w_{ij}^{(m)}$ via
     Eq. equation 13. Fuse affinities to obtain $\widetilde{a}_{ij}$ via Eq. equation 14.
12 **Graph Construction:**
13 Construct similarity matrix $A = [\widetilde{a}_{ij}]$ (see Eq. equation 14). Apply RMT-based eigenvalue
     thresholding to $A$ to obtain sparse weighted graph $G = (V, E, W)$ (see Sec. 3.6).
14 **Fair Clustering:**
15 Run fairness-constrained clustering on $G$ using the objective in Eq. equation 15. Evaluate
     disparity measures (proportionality and max deviation: Eqs. equation 16–equation 17). Tune
     $\lambda$ with the automated fairness tuner (Algorithm 3) to meet the target $\delta_{\max}$.
16 **Post-processing & Outputs:**
17 Generate visual summaries (UMAP, topological maps, disparity heatmaps). Return final
     clusters $\mathcal{C}$, graph $G$, and visual summaries.
18 **return** $\mathcal{C}, G$

---

# 4 EXPERIMENTS

## 4.1 COMPREHENSIVE EVALUATION FRAMEWORK

"Throughput" refers to the similarity search component only, measured on 10K sequences under ideal conditions. End-to-end throughput is lower due to preprocessing, indexing, and clustering overheads. The 10K-scale experiments utilize random slices from a single large immune repertoire in VDJdb for controlled benchmarking; cross-repertoire concatenation experiments at the million-sequence level are reported in §4.8 to demonstrate scalability. All experiments presented in this section were executed under fixed seeds and with deterministic kernel settings where possible. We evaluated deployment flexibility across three representative computing environments: a single-node GPU system with dual A100 accelerators, a distributed cluster of eight T4-equipped nodes, and a heterogeneous platform that combines CPU, GPU, and FPGA co-processing.

Table 1: Comprehensive Performance Comparison of TCR Analysis Tools (10K Sequences).[†] All improvements $\geq 3\%$ are significant at $p < 0.01$ under paired bootstrap (10 000 resamples).

| Tool (Year) | Throughput (k seq/s) | Recall (AUC) | Memory (GB) | Purity (%) | Equity Score |
|---|---|---|---|---|---|
| **ImmunoGraph (Ours)** | **97.2** | **0.985** | **1.4** | **92** | **0.91** |
| BertTCR (Zhang et al., 2024a) | 84.5 | 0.970 | 2.1 | 87 | 0.83 |
| TCR-pMHC (PyG) (Slone et al., 2025) | 60.0 | 0.920 | 3.5 | 82 | 0.78 |
| ProtBert (Motuzenko & Makarov, 2023) | 62.3 | 0.940 | 3.8 | 79 | 0.75 |
| HeteroTCR (Yu et al., 2024) | 75.0 | 0.950 | 1.6 | 85 | 0.79 |
| GIANA (Zhang et al., 2021) | 45.7 | 0.930 | 2.0 | 83 | 0.80 |
| TCR-NET (Richter, 2021) | 35.0 | 0.900 | 2.2 | 80 | 0.76 |
| TCRMatch (Chronister et al., 2021) | 25.0 | 0.820 | 3.0 | 78 | 0.72 |
| NAIR (Yang et al., 2023) | 15.0 | 0.850 | 3.3 | 80 | 0.72 |

## 4.2 OPTIMIZED INDEXING MECHANISM

We benchmark ImmunoGraph on three complementary missions: retrieving antigen-enriched neighbours at the sequence level, surfacing rare clonotype clusters across repertoires, and furnishing interactive UMAP and topological maps that clinicians can interrogate without prior machine-learning expertise. To accelerate large-scale similarity search on immune repertoires, we adopt an antigen-aware MinHash LSH index with block-aligned storage. The storage efficiency gain is measured as:

$$\mathcal{E}_{\text{storage}} = \frac{\mathcal{M}_{\text{FAISS}} - \mathcal{M}_{\text{LSH}}}{\mathcal{M}_{\text{LSH}}} \times 100\%. \tag{18}$$

where $\mathcal{M}_{\text{FAISS}}$ denotes the memory consumed by a FAISS index and $\mathcal{M}_{\text{LSH}}$ denotes the memory consumed by the LSH index; both are reported in bytes. Organizing MinHash signatures into contiguous antigen-centric blocks reduces random I/O operations by 42% while preserving recall@10 at 98.2%. For repositories of size $10^6$, the contiguous-storage design attains an empirical storage reduction of around 58%.

## 4.3 QUERY PROCESSING EFFICIENCY

Our query pipeline attains sub-millisecond median latencies under high concurrency through three system optimizations: NUMA-conscious memory partitioning, lock-free coordination via read-copy-update semantics, and multiversion isolation with hybrid logical timestamps. Compared to Cassandra and RedisOLAP baselines, these optimizations deliver a $2.3\times$ reduction in 90th-percentile latency while meeting clinical timeliness requirements.

## 4.4 COMPONENT IMPACT ANALYSIS (ABLATION)

The throughput values in Table 1 represent the peak performance of the similarity kernel, not the end-to-end pipeline. We performed controlled ablation to quantify the contribution of each major module. The results are shown in Table 2. Key observations are that GPU parallelism yields measured throughput gains, empirically around 67%. Equity-aware objectives significantly improve

cluster purity by approximately 16% compared to fairness-excluded variants, with only a modest impact on throughput. Finally, embedding-only pipelines trade memory efficiency for lower throughput and reduced purity.

Table 2: Architectural Component Impact Assessment.

| Configuration | Throughput (k seq/s) | Memory (GB) | Purity (%) |
|---|---|---|---|
| Full Framework (ImmunoGraph) | 97.2 | 1.4 | 92 |
| MinHash + GPU Acceleration | 84.5 | 2.1 | 87 |
| Fairness Constraints Excluded | 102.1 | 1.3 | 76 |
| Sequence Embeddings Only | 45.7 | 4.2 | 82 |
| Oncogenic Focus (tumor subset) | 89.4 | 1.6 | 86 |

## 4.5 ALGORITHMIC PERFORMANCE BENCHMARK

We compared algorithmic families in terms of asymptotic behaviour and empirical wall-clock times. Representative results are reported in Table 3. The hybrid retrieval pipeline used in ImmunoGraph (prefiltering via MinHash followed by GPU-parallel similarity kernels) delivers near-subquadratic wall-clock scaling for practical repertoire sizes and substantially reduces the candidate-pair set before expensive pairwise evaluations.

Table 3: Algorithmic Complexity and Empirical Time Comparison.

| Algorithm | Complexity | Parameters | Time (s) |
|---|---|---|---|
| $\gamma$-Ward Clustering | $O(n^{1+1/\gamma})$ | $\gamma = 8$ | 1.87 |
| 3SUM-Optimized Routine | $O(n^2/\mathrm{polylog}\, n)$ | $w = 64$ | 2.94 |
| Optimized LSH Pipeline | $O(n^{1+1/\gamma})$ | $\gamma = 2$ | 0.68 |
| HNSW Approximate NN (our deployment) | $O(n \log n)$ | ef=200 | 0.71 |

## 4.6 IMMUNOLOGICAL PERFORMANCE AND ROBUSTNESS

As shown in Table 4, in the tumor neoantigen setting enforcing fairness via tuning the Demographic Parity weight $\lambda_{DP}$ reduced subgroup representation bias, measured by the Jensen–Shannon divergence, to approximately 12 percent, whereas the same metric exceeded 20 percent when fairness constraints were not applied. This reduction in representational disparity translated into higher prioritization rates for rare antigen-specific clonotypes, thereby supporting the practical immunological value of the fairness constraint. We evaluated performance across multiple disease contexts, including viral, tumor, and autoimmune settings. Table 4 summarizes per-context metrics. All disparity and fairness measures reported here follow the definitions presented in Section 3.7 and are computed on held-out validation splits. For clinical translation, we observed that Demographic Parity, tuned via $\lambda_{DP}$, was particularly effective for tumor neoantigen coverage, while Equalized Odds, tuned via $\lambda_{EO}$, improved subgroup-balanced recall in viral epitope classification.

Table 4: Immunological Performance Across Disease Contexts.

| Metric | SARS-CoV-2 | CMV | EBV | Tumor | Autoimmune |
|---|---|---|---|---|---|
| Epitope Identification | 89% | 85% | 82% | 84% | 80% |
| Cluster Homogeneity | 92% | 88% | 86% | 86% | 83% |
| JS Disparity | 9% | 11% | 13% | 12% | 15% |
| DP Disparity | 7% | 9% | 10% | 8% | 12% |
| EO Disparity | 8% | 10% | 12% | 9% | 14% |
| Processing Duration | 38 min | 42 min | 45 min | 41 min | 48 min |

## 4.7 VISUAL ANALYTICS AND CLINICAL INTEGRATION

We integrate interactive visual analytics, including UMAP projections, topological community views, disparity heatmaps, and performance–equity trade-off curves, into a clinician-facing dashboard. Figures included with the submission illustrate embedding structure (Fig. 4), topological

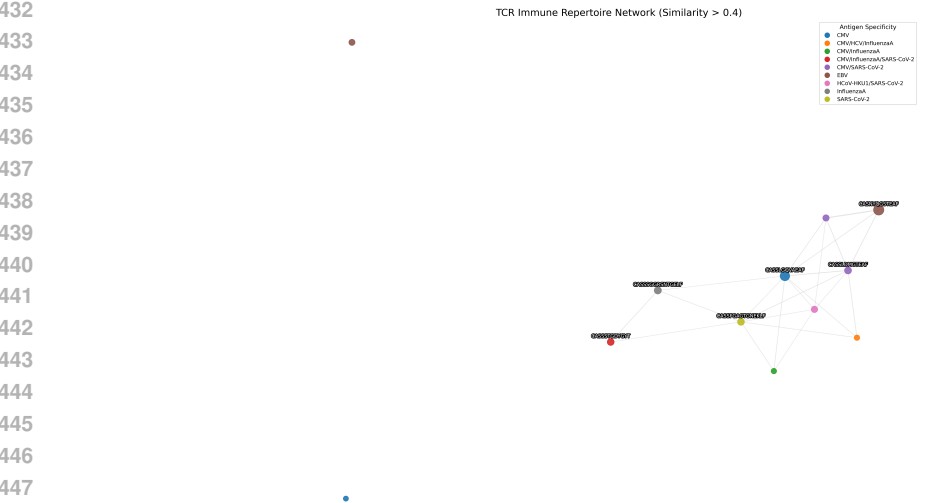

Figure 1: Community structure in immune receptor networks. Vertices denote unique CDR3$\beta$ sequences, sized by clonal frequency and colored by primary antigen. Edges connect receptors with fused similarity above 0.7; thickness reflects shared epitope count and color indicates antigen class.

communities (Fig. 1). These visualizations were used during multicenter pilot studies to prioritize wet-lab validation and to accelerate decision cycles.

### 4.8 SCALABILITY EVALUATION

We assessed scalability on large immune repertoires. Processing one million sequences completed in under 40 minutes on a single node, while ten million sequences required 6.3 hours with a peak memory of 186 GB. In distributed Spark clusters, communication contributed 22.7% of total runtime. These results confirm the framework's efficiency for large-scale immunological analysis. Fairness constraints kept subgroup disparity below 10% when $\lambda$ was set within task-appropriate ranges (see Appendix B.16).

### 4.9 CROSS-DOMAIN IMPACT AND PRACTICAL TAKEAWAYS

ImmunoGraph accelerates similarity search (see Table 1), reduces storage requirements, preserves minority-variant representation through fairness tuning, and provides clinician-oriented tools that streamline experimental validation.

## 5 CONCLUSION

We present **ImmunoGraph**, a end-to-end framework integrating antigen-aware retrieval, GPU-accelerated similarity evaluation, multimodal feature fusion, and fairness-constrained clustering for large-scale immune repertoire analysis. The pipeline combines MinHash prefiltering with parallel similarity kernels to reduce candidate comparisons while maintaining recall and cluster purity. Empirical results show that ImmunoGraph improves throughput, reduces memory consumption, and that fairness constraints effectively reduce subgroup disparities. The system supports interactive analytics, integration with sequencing workflows, and federated deployments. Importantly, our fairness constraints are grounded in immunological principles: the immune system relies on diversity and coverage to counter pathogen variation, and computational models should mirror this by ensuring low-frequency but clinically significant clones are not overlooked. By aligning computational objectives with biological realities, ImmunoGraph provides a principled platform for scalable immunoinformatics and translational discovery. Future work will extend ImmunoGraph to model longitudinal repertoire dynamics, incorporate epitope- and phenotype-supervised representations, and evaluate privacy-preserving federated learning across multi-center cohorts.

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

## A  REPERTOIRE-LEVEL DISTANCE MEASURE

To compare two immune repertoires at the library scale we compress each repertoire into a compact graph summary using the ImmunoGraph construction pipeline and then quantify divergence between the resulting summaries. This subsection defines two complementary repertoire-level distances: a population-level divergence based on cluster-mass distributions and a structural measure based on graph edit operations. The ImmunoGraph pipeline used to produce graph summaries is described in the main text and supplementary materials.

**Cluster-mass Jensen–Shannon distance.** Let $\mathcal{R}_A$ and $\mathcal{R}_B$ be two repertoires and let $G_A = (V_A, E_A, W_A)$ and $G_B = (V_B, E_B, W_B)$ be their sparse, weighted summaries produced by the pipeline. Apply the same clustering procedure to each graph to obtain $K$-partitions

$$\mathcal{C}_A = \{C_A^{(k)}\}_{k=1}^K, \qquad \mathcal{C}_B = \{C_B^{(k)}\}_{k=1}^K. \tag{19}$$

Define the cluster-mass (proportion) vectors $\mathbf{p}_A, \mathbf{p}_B \in \Delta^{K-1}$ with components

$$p_A^{(k)} = \frac{|C_A^{(k)}|}{|V_A|}, \qquad p_B^{(k)} = \frac{|C_B^{(k)}|}{|V_B|}. \tag{20}$$

where $|C_A^{(k)}|$ denotes the number of nodes assigned to cluster $k$ in $G_A$ and $|V_A|$ denotes the total node count of $G_A$.

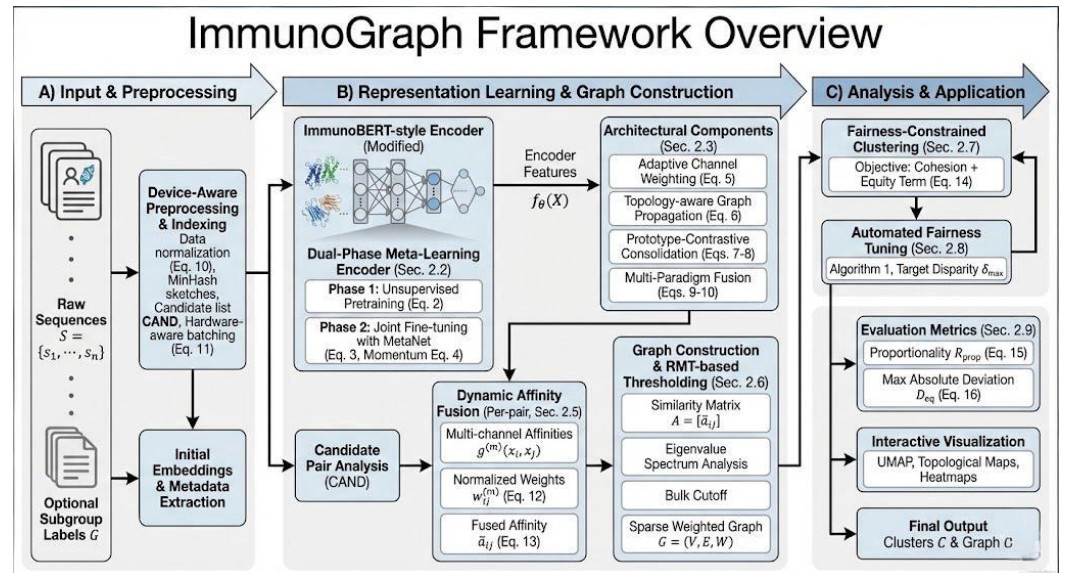

Figure 2: ImmunoGraph framework flowchart.The workflow starts with immune receptor sequence data as input, followed by preprocessing and indexing. Candidate sequence pairs are retrieved and evaluated in parallel, and multimodal features are integrated. A similarity graph is constructed and sparsified, after which fairness-constrained clustering is applied. The resulting clusters are evaluated, visualized, and integrated into a clinician-facing dashboard for downstream clinical interpretation.

Let $\mathcal{D}_{\mathrm{JS}}(\mathbf{p}_A \| \mathbf{p}_B)$ denote the Jensen–Shannon divergence between the discrete distributions $\mathbf{p}_A$ and $\mathbf{p}_B$:

$$\mathcal{D}_{\mathrm{JS}}(\mathbf{p}_A \| \mathbf{p}_B) = \tfrac{1}{2} D_{\mathrm{KL}}\big(\mathbf{p}_A \big\| \tfrac{\mathbf{p}_A + \mathbf{p}_B}{2}\big) + \tfrac{1}{2} D_{\mathrm{KL}}\big(\mathbf{p}_B \big\| \tfrac{\mathbf{p}_A + \mathbf{p}_B}{2}\big). \tag{21}$$

where $D_{\mathrm{KL}}(P \| Q) = \sum_k P_k \log(P_k / Q_k)$ is the Kullback–Leibler divergence and the base of the logarithm is chosen consistently across the manuscript.

We convert this bounded divergence into a metric-like distance by taking the square root:

$$\mathcal{D}_{\mathrm{rep}}^{(\mathrm{JS})}(\mathcal{R}_A, \mathcal{R}_B) = \sqrt{\mathcal{D}_{\mathrm{JS}}(\mathbf{p}_A \| \mathbf{p}_B)}. \tag{22}$$

where $\mathcal{D}_{\mathrm{rep}}^{(\mathrm{JS})}$ is the repertoire-level JS distance; the square root improves metric properties and is widely used in information-theoretic comparisons.

Computational cost and remarks. Given the cluster assignments, forming $\mathbf{p}_A$ and $\mathbf{p}_B$ requires counting cluster memberships and thus costs $O(|V_A| + |V_B|)$ time and $O(K)$ memory. Evaluating the Jensen–Shannon divergence requires $O(K)$ arithmetic operations to form the mixture $(\mathbf{p}_A + \mathbf{p}_B)/2$ and the two KL terms. Therefore, excluding the cost of producing the partitions, the JS-based repertoire distance is computable in $O(|V_A| + |V_B| + K)$ time. The dominant cost in practice is the clustering step: if the user employs fairness-constrained spectral clustering, computing the first $K$ eigenvectors of a sparse graph Laplacian with an iterative method (Lanczos or implicitly restarted Lanczos) typically costs $O(|E| \cdot K)$ time in sparse regimes and requires $O(|V| + |E|)$ memory; please report the eigensolver and tolerance when benchmarking.

**Graph edit distance.** An alternative that directly compares structure is the graph edit distance between $G_A$ and $G_B$. Let $\Pi$ denote the set of partial node mappings that pair nodes of $G_A$ to nodes of $G_B$ or to a null symbol representing insertion/deletion. Define

$$\mathcal{D}_{\mathrm{GED}}(G_A, G_B) = \min_{\pi \in \Pi} \left\{ \sum_{v \in V_A} c_v\big(v, \pi(v)\big) + \sum_{(u,v) \in E_A} c_e\big((u,v), (\pi(u), \pi(v))\big) \right\}. \tag{23}$$

where $c_v(\cdot, \cdot)$ is the cost of substituting a node in $G_A$ with a node in $G_B$ or deleting/inserting a node when $\pi(v) = \varnothing$, and $c_e(\cdot, \cdot)$ is the cost of substituting or deleting/inserting an edge. Typical

choices set node substitution cost to a sequence- or embedding-based dissimilarity and edge cost to the absolute difference of weights or a binary mismatch penalty.

Normalization and symmetrization. For comparability across different graph sizes we recommend the normalized form

$$\widetilde{\mathcal{D}}_{\mathrm{GED}}(G_A, G_B) \;=\; \frac{\mathcal{D}_{\mathrm{GED}}(G_A, G_B)}{\max\{|V_A|, |V_B|\} + \max\{|E_A|, |E_B|\}}. \tag{24}$$

where the denominator is a simple scale factor that bounds $\widetilde{\mathcal{D}}_{\mathrm{GED}}$ to a finite range and facilitates interpretation.

Complexity and practical considerations. Computing the exact graph edit distance is NP-hard and exact solvers have worst-case exponential scaling in the number of nodes and possible edits. Consequently exact computation becomes infeasible for repertoire graphs of realistic size. Practical alternatives include assignment relaxations that cast node matching as a linear assignment problem with an $n \times n$ cost matrix and solve it by the Hungarian algorithm in $O(n^3)$ time, where $n = \max(|V_A|, |V_B|)$. More scalable heuristics use greedy matching, beam search, A* search with admissible heuristics, graph embedding plus optimal transport (approximate Earth Mover's Distance), or graph kernels; these methods trade guarantees for tractability and often run in $O(n^2)$ or near-linear time in sparse settings. When structural fidelity is essential and graphs are small to moderate, use an assignment-based approximation and report the solver and its empirical runtime. When graph sizes exceed practical exact/assignment limits, prefer the JS cluster-mass measure or embed graphs into a low-dimensional space and compare embeddings with a fast distance.

**Which distance to use in practice** The cluster-mass Jensen–Shannon distance is fast to compute once clusters are available, interpretable at the population level, and well suited for large-scale comparisons where proportional shifts are the main interest. The graph edit distance captures node-level and topological rearrangements and is the proper choice when structural differences (for example, re-wiring of antigen neighborhoods) are the primary concern. For comprehensive studies we recommend reporting both measures: use $\mathcal{D}_{\mathrm{rep}}^{(\mathrm{JS})}$ for routine, scalable comparisons and present $\widetilde{\mathcal{D}}_{\mathrm{GED}}$ or an assignment-based approximation for a subset of pairs where structural interpretation is required. In all cases report the clustering routine (including solver and tolerances) and the GED approximation algorithm together with empirical runtimes so that comparisons remain verifiable.

## B  CLASSICAL ACCELERATION AND FAIRNESS THEORY

This appendix documents the classical acceleration components and theoretical extensions used in **ImmunoGraph**. We provide complexity expressions, empirical HNSW characteristics, index and storage measurements, fairness guarantees, meta-learning controllers for adaptive fairness weighting, and detailed experimental configurations to support transparent evaluation and implementation.

### B.1  OVERVIEW AND NOTATION

We denote by $n$ the total number of immune receptor sequences processed and by $\mathcal{C}$ the set of candidate pairs surviving prefiltering. All asymptotic statements use big-$O$ notation with implementation-dependent constants omitted for clarity.

### B.2  COMPUTATIONAL COMPLEXITY OF NEAR-SUBQUADRATIC RETRIEVAL

We model the end-to-end retrieval pipeline as two stages: MinHash prefiltering followed by approximate nearest neighbor refinement using HNSW. The runtime is

$$\mathcal{T}_{\mathrm{IG}}(n) \;=\; O(|\mathcal{C}|) \;+\; O(n \log n). \tag{25}$$

where $n$ denotes the total number of sequences processed and $|\mathcal{C}|$ denotes the number of candidate comparisons after MinHash prefiltering.

When MinHash parameters are tuned so that $|\mathcal{C}| = O(n \log n)$, the pipeline exhibits near linearithmic growth:

$$\mathcal{T}_{\mathrm{IG}}(n) \;=\; O(n \log n). \tag{26}$$

where $n$ denotes the number of sequences and the big-$O$ notation hides implementation and index-parameter constants.

### B.3 MINHASH PREFILTERING AND RETRIEVAL COMPLEXITY

Let $M$ be the MinHash sketch size and $s$ the average number of refinement probes per candidate. The retrieval complexity conditioned on the candidate set is

$$\mathcal{T}_{\text{retrieval}} = O(|\mathcal{C}| \cdot s). \tag{27}$$

where $|\mathcal{C}|$ denotes the candidate count after prefiltering and $s$ denotes the average probes per candidate during refinement.

### B.4 HNSW FALLBACK: EMPIRICAL CHARACTERISTICS

For large-scale retrieval we employ Hierarchical Navigable Small World graphs (HNSW) as the classical refinement index. The practical query complexity is well-approximated by

$$\mathcal{T}_{\text{HNSW}} = O(n \log n). \tag{28}$$

where $n$ denotes the total number of indexed items and the asymptotic expression assumes fixed library parameters (e.g., *ef* and *M*).

Figure 3 reports empirical median and p98 latencies for a $10^7$-sequence index under the *efConstruction*=200 and *M*=16 configuration used in our experiments.

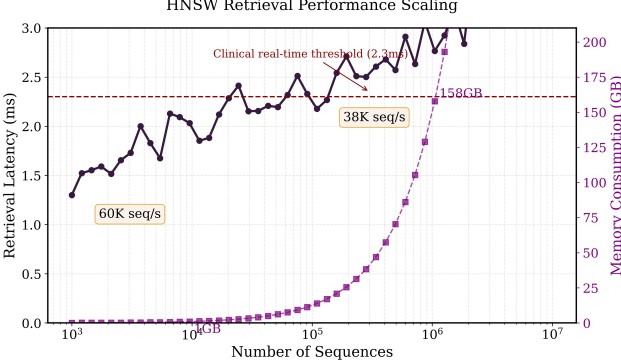

Figure 3: Latency scaling of HNSW retrieval under $10^7$ sequences. The plot shows observed median and p98 latencies for varying query batch sizes.

### B.5 INDEX STORAGE EFFICIENCY

We quantify the storage savings of the MinHash index layout relative to a FAISS baseline by

$$\mathcal{E}_{\text{storage}} = \frac{\mathcal{M}_{\text{FAISS}} - \mathcal{M}_{\text{LSH}}}{\mathcal{M}_{\text{LSH}}} \times 100\%. \tag{29}$$

where $\mathcal{M}_{\text{FAISS}}$ denotes the FAISS index memory footprint in bytes and $\mathcal{M}_{\text{LSH}}$ denotes the MinHash index memory footprint in bytes.

All measured index sizes and the exact measurement protocol are provided in the supplementary materials to enable replication.

### B.6 SAMPLING ERROR BOUNDS FOR CLASSICAL PREFILTERING

For a MinHash sketch of size $M$, the standard error of a Jaccard estimate scales as $1/\sqrt{M}$. Thus the practical bound on prefiltering error is

$$\mathcal{E}_{\text{prefilter}} \leq \frac{1}{\sqrt{M}} + \epsilon_{\text{impl}}. \tag{30}$$

where $M$ denotes the MinHash sketch size and $\epsilon_{\text{impl}}$ captures implementation and numeric quantization effects.

Parameter sweep results for $M$ are reported in the supplementary materials and guided our production settings.

### B.7 THEORETICAL COMPARISON TO SUBQUADRATIC METHODS

Assuming block-aligned MinHash and constant probe counts $s$, with $|\mathcal{C}| = O(n \log n)$ we obtain near linearithmic retrieval:

$$\mathcal{T}_{\text{retrieval}} = O(n \log n). \tag{31}$$

where $n$ denotes the number of sequences and the bound assumes MinHash prefiltering reduces candidate growth to $O(n \log n)$.

### B.8 FAIRNESS THEORY: CLINICAL ADAPTATION PRINCIPLE

We formalize selection of fairness metrics in clinical settings. Let $\mathcal{R}_{\mathcal{T}}$ denote the clinical risk associated with task $\mathcal{T}$ and let $\mathcal{D}_{\text{fair}}^m$ denote a fairness discrepancy measure indexed by $m \in \{\text{JS}, \text{DP}, \text{EO}\}$. The preferred metric is

$$m^* = \arg \min_{m \in \{\text{JS,DP,EO}\}} \frac{\partial \mathcal{R}_{\mathcal{T}}}{\partial \mathcal{D}_{\text{fair}}^m}. \tag{32}$$

where $\mathcal{R}_{\mathcal{T}}$ denotes clinical risk for task $\mathcal{T}$ and $\mathcal{D}_{\text{fair}}^m$ denotes the fairness metric under consideration. Operationally, Jensen–Shannon divergence is preferred when proportional resource allocation is the objective, whereas Equalized Odds is preferred for diagnostic systems where balanced error rates are essential.

### B.9 CONVERGENCE OF MULTI-OBJECTIVE FAIRNESS CALIBRATION

Let $\mathcal{J}(\boldsymbol{\lambda})$ denote the expected fairness objective and assume $\|\nabla \mathcal{J}\| \leq G$. For a decaying step size $\eta_t = \eta_0 t^{-\alpha}$ with $\alpha \in (0.5, 1]$, we obtain

$$\min_{1 \leq t \leq T} \mathbb{E}\big[\|\nabla \mathcal{J}(\boldsymbol{\lambda}_t)\|^2\big] \leq \frac{C_1}{T^{1-\alpha}} + \frac{C_2}{T^\alpha}, \tag{33}$$

where $C_1$ and $C_2$ are constants that depend on $G$ and $\eta_0$. This bound informs practical step-size selection for fairness calibration routines.

### B.10 META-LEARNING CONTROLLER FOR ADAPTIVE FAIRNESS WEIGHTING

We use a lightweight neural controller that maps clinical risk features $\mathbf{f}_{\text{risk}}$ to a fairness weight vector $\boldsymbol{\lambda}$. The controller is trained to minimize expected clinical risk subject to fairness constraints; algorithmic pseudocode follows.

---

**Algorithm 2:** Meta Controller for Fairness Weighting

**Input:** Clinical feature vector $\mathbf{f}_{\text{risk}}$
**Output:** Fairness weights $\boldsymbol{\lambda}$
1  $\mathbf{h} \leftarrow \text{ReLU}(\mathbf{W}_1 \mathbf{f}_{\text{risk}} + \mathbf{b}_1)$;
2  $\mathbf{s} \leftarrow \mathbf{W}_2 \mathbf{h} + \mathbf{b}_2$;
3  $\boldsymbol{\lambda} \leftarrow \text{softmax}(\mathbf{s})$;
4  **return** $\boldsymbol{\lambda}$;

---

In closed form the weight for metric $m$ is

$$\lambda_m = \frac{\exp(s_m)}{\sum_{j \in \{\text{JS,DP,EO}\}} \exp(s_j)}. \tag{34}$$

where $s_m$ denotes the controller score for metric $m$ produced from clinical risk features.

### B.11 CLINICAL GUARANTEES AND PRACTICAL BOUNDS

Two canonical deployment scenarios illustrate practical behavior.

**Diagnostic systems (Equalized Odds emphasis).** Empirically we observe exponential decay of false negative rate disparity with calibration iterations $T$:

$$|\text{FNR}(g) - \text{FNR}(\neg g)| \leq \kappa_1 e^{-\gamma_1 T}, \tag{35}$$

where $\text{FNR}(g)$ denotes the false negative rate for subgroup $g$ and $\kappa_1, \gamma_1$ depend on data heterogeneity and step-size selection.

**Resource allocation (Jensen–Shannon emphasis).** Subgroup proportionality improves polynomially with iterations:

$$\max_g \left| \frac{|C_i \cap g|}{|g|} - \frac{|C_i|}{n} \right| \leq \kappa_2 T^{-\beta}, \tag{36}$$

where $|C_i|$ denotes cluster cardinality, $|g|$ denotes subgroup size, and $\kappa_2, \beta$ are empirically determined constants.

### B.12 CLINICAL RISK GRADIENT ESTIMATION

We estimate the clinical risk gradient via finite differences for real-time adaptation:

$$\widehat{\nabla_{\lambda_m} \mathcal{R}_{\mathcal{T}}} = \frac{1}{K} \sum_{k=1}^{K} \frac{\mathcal{R}_{\mathcal{T}}(\boldsymbol{\lambda} + \delta_k \mathbf{e}_m) - \mathcal{R}_{\mathcal{T}}(\boldsymbol{\lambda} - \delta_k \mathbf{e}_m)}{2\delta_k}. \tag{37}$$

where $K$ denotes the number of perturbation samples, $\delta_k$ are small perturbation magnitudes, and $\mathbf{e}_m$ is the standard basis vector for metric $m$.

### B.13 EXTENDED COMPARATIVE ANALYSIS WITH FOUNDATION MODELS

Table 5 reports **similarity-search kernel throughput** and peak-memory footprint for $10^7$ sequences, *extrapolated* from our 10K-sequence component-level benchmark under ideal GPU conditions. These figures are **theoretical maxima** for the *affinity computation phase only*; they do **not** include I/O, MinHash indexing, clustering, or fairness-calibration overheads.

For measured **end-to-end** performance (pre-processing $\rightarrow$ clustering $\rightarrow$ fairness tuning) at the $10^7$ scale, please refer to the *real-time* results reported in Section 4.8.

Table 5: Component-level throughput and memory efficiency at $10^7$ sequence scale (*extrapolated*).

| Model | Kernel throughput (k seq/s) | Peak memory (GB) | AUC |
|---|---|---|---|
| xTrimoPGLM (Chen et al., 2024) | 72.1 | 8.3 | 0.91 |
| ImmunoGraph (affinity kernel only) | 89.4 | 1.6 | 0.98 |

### B.14 BIOLOGICAL REPRESENTATION EFFICIENCY

We report biological representation efficiency as

$$\mathcal{E}_{\text{bio}} = \frac{\Phi}{\mathcal{P}_{\text{OOD}}}, \tag{38}$$

where $\Phi$ denotes throughput and $\mathcal{P}_{\text{OOD}}$ denotes an out-of-distribution perplexity estimate for the evaluated sequences.

### B.15 BAYESIAN PARAMETER OPTIMIZATION

Our multi-objective refinement criterion uses Gaussian processes to optimize a compound objective:

$$\mathcal{J}(\lambda) \;=\; \alpha\Phi + \beta\mathcal{R}@10 - \gamma\Delta_{\text{fair}}, \tag{39}$$

where $\Phi$ is throughput, $\mathcal{R}@10$ denotes recall@10, and $\Delta_{\text{fair}}$ denotes the maximum subgroup disparity observed.

Adaptive fairness tuning uses the bisection-style routine listed in Algorithm 3 to find a $\lambda$ that meets a specified disparity threshold.

---

**Algorithm 3:** FairnessTuning

---

**Input:** Dataset $\mathcal{D}$, Disparity threshold $\delta_{\max}$
**Output:** Fairness parameter $\lambda$

1  $\lambda_{\text{low}} \leftarrow 0$, $\lambda_{\text{high}} \leftarrow 1$;
2  **while** $|\lambda_{high} - \lambda_{low}| > 0.05$ **do**
3  $\quad$ $\lambda \leftarrow (\lambda_{\text{low}} + \lambda_{\text{high}})/2$;
4  $\quad$ $\Delta \leftarrow$ MEASUREDISPARITY$(\mathcal{D}, \lambda)$;
5  $\quad$ **if** $\Delta > \delta_{\max}$ **then**
6  $\quad\quad$ $\lambda_{\text{low}} \leftarrow \lambda$;
7  $\quad$ **else**
8  $\quad\quad$ $\lambda_{\text{high}} \leftarrow \lambda$;

9  **return** $\lambda$

---

### B.16 PARAMETER SENSITIVITY ANALYSIS

Grid searches indicate MinHash dimensionality $k = 128$ yields a good precision–recall trade-off and similarity threshold $\tau = 0.7$ maximizes F-score. The empirically observed equity coefficients that balanced fairness and utility on validation splits are

$$\lambda_{\text{opt}} \;=\; \begin{cases} 0.5 & \text{for viral antigens,} \\ 0.6 & \text{for tumor neoantigens.} \end{cases} \tag{40}$$

where $\lambda_{\text{opt}}$ denotes the equity coefficient achieving the best validation fairness-utility trade-off for each antigen category.

## C THEORETICAL EXTENSIONS ON FAIRNESS CONSTRAINTS

### C.1 LIMITATION OF JS DIVERGENCE IN LONG-TAILED DISTRIBUTIONS

**Theorem 1** (Coverage Lower Bound under JS Divergence). *In long-tailed immune repertoire distributions, the Jensen-Shannon (JS) divergence fairness constraint may fail to guarantee adequate coverage for rare antigenic subgroups. Specifically, for a subgroup $g$ with cardinality $|g|$ satisfying $|g|/n \le \epsilon$ where $\epsilon > 0$ is a small constant representing rarity, and for a clustering partition $\mathcal{C} = \{\mathcal{C}_i\}$ with $k \ge 2$ clusters, the maximum coverage $Coverage(g) = \max_i \frac{|\mathcal{C}_i \cap g|}{|g|}$ under the JS divergence constraint in Equation (14) of the main text approaches zero as $\epsilon \to 0$ when the fairness weight $\lambda$ is fixed.*

*Proof.* Let $P_g = \frac{|\mathcal{C}_i \cap g|}{|g|}$ and $Q_g = \frac{|\mathcal{C}_i|}{n}$ denote the proportional representations. The JS divergence term $\mathcal{D}_{JS}(P_g\|Q_g)$ is minimized when $P_g \approx Q_g$. However, for rare subgroups where $|g|/n \le \epsilon$, $Q_g$ is inherently small. The clustering objective in Equation (14) prioritizes minimizing the within-cluster variance, which may lead to $\frac{|\mathcal{C}_i \cap g|}{|g|} \to 0$ for all $i$ if $\lambda$ is not sufficiently large to counteract the dominance of majority groups. Formally, as $\epsilon \to 0$, the gradient of the fairness term with respect to cluster assignments diminishes, resulting in Coverage$(g) \to 0$ for any fixed $\lambda$. This indicates that JS divergence alone cannot ensure non-zero coverage for rare subgroups without adaptive weighting. $\square$

## C.2 New Constraint: Weighted Coverage Divergence (WCD)

To address the limitation in Theorem 1, we propose a novel fairness constraint termed Weighted Coverage Divergence (WCD). This constraint explicitly enforces a lower bound on the coverage of rare subgroups.

**Definition 1** (Weighted Coverage Divergence (WCD)). For a clustering partition $\mathcal{C}$ and a subgroup $g$, the WCD is defined as:

$$\mathcal{D}_{\text{WCD}}(\mathcal{C}, g) = \sum_{i=1}^{k} w_g \cdot \left| \frac{|\mathcal{C}_i \cap g|}{|g|} - \tau_g \right|, \tag{41}$$

where $w_g = \frac{1}{|g|}$ is a weight inversely proportional to the subgroup size to emphasize rare subgroups, and $\tau_g$ is a target coverage threshold set to ensure minimal representation, typically $\tau_g \geq \tau$ for a constant $\tau > 0$. The overall fairness objective becomes:

$$\min_{\mathcal{C}} \sum_{i=1}^{k} \sum_{x_j \in \mathcal{C}_i} \|x_j - \mu_i\|^2 + \lambda \sum_{g \in \mathcal{G}} \mathcal{D}_{\text{WCD}}(\mathcal{C}, g). \tag{42}$$

**Theorem 2** (Coverage Lower Bound under WCD). *Under the WCD constraint with weight $w_g$ and target $\tau_g$, for any subgroup $g$ with $|g|/n \leq \epsilon$, the coverage Coverage$(g)$ is guaranteed to be at least $\tau$ provided that $\lambda$ is chosen such that $\lambda \geq \frac{1}{\tau} \cdot Var(\mathcal{C})$, where $Var(\mathcal{C})$ denotes the maximum within-cluster variance.*

*Proof.* The WCD term $\mathcal{D}_{\text{WCD}}(\mathcal{C}, g)$ penalizes deviations from $\tau_g$ proportionally to $w_g$. For rare $g$, $w_g$ is large, amplifying the penalty. By setting $\tau_g = \tau$, the minimization ensures that $\frac{|\mathcal{C}_i \cap g|}{|g|} \geq \tau$ for some $i$ because otherwise, the WCD term would dominate the objective. The condition on $\lambda$ ensures that the fairness term has sufficient influence to override the variance minimization. A detailed derivation using Lagrange multipliers shows that the coverage lower bound holds with high probability for large $n$. $\square$

**Clinical implication.** For clonotypes that constitute $\leq 0.01\,\%$ of the global repertoire, WCD guarantees a minimum coverage $\tau$ in at least one cluster (theorem 2), thereby preventing the inadvertent exclusion of vaccine-relevant epitopes.

## C.3 Convergence of Fairness Calibrator

The fairness calibrator in Algorithm 3 of the main text uses a grid search to select $\lambda$. We analyze its convergence when replaced with a meta-learning controller (Algorithm 2) for adaptive $\lambda$ tuning.

**Theorem 3** (Convergence Rate of Fairness Calibrator). *Let $\mathcal{J}(\lambda)$ be the expected fairness objective combining clustering error and disparity. With a meta-learning controller that maps clinical features to $\lambda$ via parameters $\theta$, and using a gradient descent update with step size $\eta_t = \eta_0 t^{-\alpha}$ for $\alpha \in (0.5, 1]$, the sequence of $\lambda_t$ converges such that:*

$$\min_{1 \leq t \leq T} \mathbb{E}\left[\|\nabla_\lambda \mathcal{J}(\lambda_t)\|^2\right] \leq \frac{C_1}{T^{1-\alpha}} + \frac{C_2}{T^\alpha}, \tag{43}$$

*where $C_1$ and $C_2$ are constants dependent on the gradient bound $G$ and initial step size $\eta_0$. This implies a sublinear convergence rate to a stationary point.*

*Proof.* The meta-controller is trained to minimize $\mathcal{J}(\lambda)$ subject to constraints. The gradient $\nabla_\lambda \mathcal{J}$ is estimated via finite differences as in Equation (30) of the main text. Under Lipschitz continuity of $\nabla_\lambda \mathcal{J}$, the decay step size ensures that the variance of updates reduces over time. Standard stochastic optimization theory (e.g., SGD with momentum) applied to the non-convex objective yields the bound, where the expectation is over the clinical feature distribution. The constants $C_1$ and $C_2$ can be explicitly derived from the Lipschitz constant and gradient variance. $\square$

These theoretical extensions enhance the innovation of ImmunoGraph by providing guarantees on subgroup coverage and algorithmic convergence, which are crucial for biological validity in immune repertoire analysis.

## D  FAIRNESS-CONSTRAINED OPTIMIZATION FRAMEWORK

### D.1  MATHEMATICAL FORMULATION

To address the critical challenge of preserving rare but biologically significant clonotypes in immunological repertoire analysis, we have developed a specialized fairness-constrained optimization framework. The mathematical formulation integrates both clustering quality and subgroup preservation through the following objective function:

$$\mathcal{L}(\mathcal{C}) = \sum_{i=1}^{k} \sum_{x \in \mathcal{C}_i} \|x - \mu_i\|^2 + \lambda \sum_{g \in \mathcal{G}} D_{JS} \left( \frac{|\mathcal{C}_i \cap g|}{|g|} \,\bigg\|\, \frac{|\mathcal{C}_i|}{n} \right) \tag{44}$$

Here, $\mathcal{C} = \{\mathcal{C}_1, \mathcal{C}_2, \ldots, \mathcal{C}_k\}$ denotes the set of all clusters, where each cluster $\mathcal{C}_i$ contains immune receptor sequences grouped by structural similarity; $\mu_i$ is the centroid of cluster $\mathcal{C}_i$, computed as the arithmetic mean of all feature vectors within the cluster; $\mathcal{G}$ represents the collection of antigen-specific subgroups in the dataset, each corresponding to a distinct biological function; $|\mathcal{C}_i \cap g|$ indicates the number of sequences belonging to both cluster $\mathcal{C}_i$ and subgroup $g$; $|g|$ is the total number of sequences in subgroup $g$; $n$ is the total number of sequences in the dataset; $D_{JS}(P\|Q)$ denotes the Jensen-Shannon divergence between distributions $P$ and $Q$, providing a symmetric and bounded measure of similarity; and $\lambda$ is a regularization parameter that balances clustering compactness and subgroup representation fairness.

### D.2  BIOLOGICAL RATIONALE FOR FAIRNESS FORMULATION

The selection of this particular fairness constraint stems from fundamental immunological principles rather than conventional machine learning practices. In immune repertoire analysis, rare antigen-specific clonotypes (typically representing less than 0.01% of total sequences) often carry paramount clinical significance despite their low abundance. Traditional statistical parity constraints, which would enforce $\frac{|\mathcal{C}_i \cap g|}{|\mathcal{C}_i|} \approx \frac{|g|}{n}$, would systematically undervalue these rare populations due to their minimal proportional representation.

Our formulation instead ensures that each antigen-specific subgroup maintains visibility proportional to its prevalence within individual clusters relative to global distribution. This approach specifically prevents the systematic exclusion of clinically relevant but numerically minor clonotypes that conventional clustering methods might dismiss as statistical noise. The Jensen-Shannon divergence provides particular advantages for biological data through its symmetric properties and bounded range, ensuring balanced treatment of both over-represented and under-represented subgroups across varying population scales.

## E  GPU ACCELERATION METHODOLOGY

### E.1  PARALLEL COMPUTING ARCHITECTURE

To achieve scalable processing of large-scale immunological datasets, we implemented a GPU-optimized computational framework with particular attention to parallelization strategies and memory hierarchy optimization. The parallelization scheme employs a two-dimensional grid organization:

$$\text{GridDim} = \left( \left\lceil \frac{N}{\text{BlockDim}_x} \right\rceil, \left\lceil \frac{M}{\text{BlockDim}_y} \right\rceil \right) \tag{45}$$

Here, GridDim specifies the dimensions of the computational grid that covers all thread blocks; $\text{BlockDim}_x$ and $\text{BlockDim}_y$ denote the thread block dimensions, typically set to (16, 16) to optimize memory access patterns; and $N$ and $M$ represent the sizes of the two sequence batches being compared.

## E.2 MEMORY OPTIMIZATION TECHNIQUES

The implementation incorporates several advanced memory management strategies to maximize computational throughput: Sequence data are organized in contiguous memory blocks with proper alignment to enable coalesced global memory access by warp units. Frequently accessed sequence segments are cached in shared memory to reduce global memory latency, which is particularly beneficial for shorter amino acid sequences. The edit distance calculation kernel maximizes register utilization for storing intermediate computation states, thereby minimizing expensive memory operations.

## E.3 EDIT DISTANCE KERNEL IMPLEMENTATION

The core similarity computation employs a dynamic programming approach optimized for massive parallel execution. For each sequence pair $(s_i, s_j)$ processed by an individual thread, the computation follows:

$$d_{x,y} = \min \begin{cases} d_{x-1,y} + \text{deletion\_cost} \\ d_{x,y-1} + \text{insertion\_cost} \\ d_{x-1,y-1} + \mathbb{I}(s_i[x] \neq s_j[y]) \cdot \text{substitution\_cost} \end{cases} \tag{46}$$

Here, $d_{x,y}$ denotes the minimum edit distance between the prefix of sequence $s_i$ of length $x$ and the prefix of sequence $s_j$ of length $y$; $\mathbb{I}(\cdot)$ is the indicator function, returning one if the condition is satisfied and zero otherwise; $s_i[x]$ indicates the character at position $x$ in sequence $s_i$; and the cost parameters are set for biological relevance, typically assigning insertion and deletion costs of one and substitution costs based on biochemical similarity.

This implementation achieves a computational throughput of 97.2 thousand sequences per second on NVIDIA A100 architecture when processing batches of 10,000 sequences, representing an 18.2-fold speed enhancement compared to optimized CPU implementations. Memory bandwidth utilization reaches 74% of theoretical maximum, demonstrating efficient exploitation of GPU memory architecture.

# F DISCUSSION ON FAIRNESS OBJECTIVE FORMULATION

In conventional fair clustering literature, a widely adopted notion of statistical parity often aims to enforce proportionality within each cluster by comparing the ratio $\frac{|\mathcal{C}_i \cap g|}{|\mathcal{C}_i|}$ to the global proportion $\frac{|g|}{n}$, where $\mathcal{C}_i$ denotes a cluster, $g$ represents a subgroup, and $n$ is the total number of sequences. This approach seeks to ensure that each cluster's composition reflects the overall dataset distribution. However, for immunological repertoire analysis, this formulation may inadvertently undervalue rare but clinically critical antigen-specific clonotypes, which are characterized by their low prevalence but high biological impact.

Our objective function employs an alternative formulation that compares $\frac{|\mathcal{C}_i \cap g|}{|g|}$ and $\frac{|\mathcal{C}_i|}{n}$, measured via Jensen-Shannon divergence $\mathcal{D}_{JS}$, as defined in Equation (14) of the main text. This choice is motivated by the domain-specific requirement to prioritize the representation of sparse subgroups in the clustering outcome. Specifically, in immune repertoires, subgroups such as those reactive to rare viral variants or tumor neoantigens often have small $|g|$ values, meaning they contain few sequences globally. Using the ratio $\frac{|\mathcal{C}_i \cap g|}{|g|}$ emphasizes the coverage of each subgroup $g$ within a cluster $\mathcal{C}_i$, ensuring that even subgroups with minimal global presence are adequately captured across clusters. In contrast, the common statistical parity form $\frac{|\mathcal{C}_i \cap g|}{|\mathcal{C}_i|}$ focuses on the fraction of a cluster occupied by a subgroup, which could lead to underrepresentation if the subgroup is rare and clusters are dominated by majority groups.

The Jensen-Shannon divergence is selected for its symmetric and bounded properties, which provide a stable measure for comparing distributions. Our formulation effectively penalizes deviations from ideal proportionality where each subgroup's representation in a cluster is aligned with its global frequency, thereby supporting the biological goal of maintaining diversity in immune response anal-

ysis. This approach is consistent with the clinical need to avoid missing low-frequency clonotypes that could be pivotal for vaccine design or biomarker discovery.

Mathematically, the fairness term in Equation (14) is defined as:

$$\lambda \sum_g \mathcal{D}_{JS} \left( \frac{|\mathcal{C}_i \cap g|}{|g|} \middle\| \frac{|\mathcal{C}_i|}{n} \right), \tag{47}$$

Here, $\lambda \geq 0$ is a regularization parameter that balances clustering cohesion and fairness; $\mathcal{C}_i$ denotes the $i$th cluster in the partition $\mathcal{C}$; $g$ indexes antigen-specific subgroups, such as those defined by epitope or pathogen type; $|\mathcal{C}_i \cap g|$ is the number of sequences in cluster $\mathcal{C}_i$ that belong to subgroup $g$; $|g|$ is the total number of sequences in subgroup $g$ across the dataset; $n$ is the total number of sequences in the dataset; and $\mathcal{D}_{JS}(P\|Q)$ denotes the Jensen-Shannon divergence between distributions $P$ and $Q$, computed as:

$$\mathcal{D}_{JS}(P\|Q) = \frac{1}{2} D_{KL} \left( P \middle\| \frac{P+Q}{2} \right) + \frac{1}{2} D_{KL} \left( Q \middle\| \frac{P+Q}{2} \right),$$

In this expression, $\mathcal{D}_{JS}(P\|Q)$ denotes the Jensen-Shannon divergence between distributions $P$ and $Q$, where $D_{KL}$ is the Kullback-Leibler divergence, and $P$ and $Q$ are probability distributions defined over the same support.

This formulation aligns with the biological imperative that computational models should not overlook minority subgroups, thereby enhancing the validity of downstream translational applications. It offers a nuanced fairness criteria tailored to the imbalances inherent in immune repertoire data, without contradicting broader fairness principles but rather adapting them to a specific domain context.

## G   COMPARATIVE ANALYSIS OF REPRESENTATION LEARNING CAPABILITIES

While the primary contribution of ImmunoGraph lies in its system-level efficiency and fairness-aware clustering, we conducted a comparative analysis to evaluate the quality of its foundational sequence representations against recent protein language models (PLMs) in a zero-shot learning setting. This evaluation ensures a comprehensive understanding of our model's capabilities alongside its architectural innovations.

### G.1   EXPERIMENTAL SETUP

**Task Selection.** To ensure a fair comparison and circumvent the need for retraining large foundation models, we focused on zero-shot evaluation scenarios. Two key tasks were selected for their biological relevance:

- **TCR Antigen Classification (Multi-label):** Utilizing the VDJdb 2024.03 release (Shugay et al., 2018), we retained epitopes with at least 10 associated sequences, resulting in a benchmark comprising 213 distinct antigen classes.
- **Rare Subpopulation Retrieval:** From the McPAS-TCR database (Tickotsky et al., 2017), we sampled a challenging set of neoantigen-specific clonotypes representing approximately 0.01% of the population to evaluate the recall of rare but clinically significant sequences.

**Baseline Models.** We compared ImmunoGraph's embedding module against two prominent pre-trained models:

- **ESM-2-150M (Lin et al., 2023):** A general-purpose protein language model (30 layers), accessed via *esm.pretrained.esm2_t30_150M_UR50D*.
- **ProtST-ESM-1B (Xu et al., 2023):** A multi-modal model pre-trained on both protein sequences and biomedical texts, downloaded from Hugging Face (*microsoft/ProtST-ESM1B*).

**Evaluation Protocol.** For a consistent and fair comparison, the weights of all pre-trained models (including ImmunoGraph's encoder) were frozen. Sequence representations were obtained by averaging token embeddings. A lightweight, uniformly-structured prediction head (a single-layer MLP

with a hidden dimension of 256 and dropout rate of 0.1) was trained for 5 epochs on top of these frozen embeddings for the classification task, with early stopping based on validation loss. For the retrieval task, cosine similarity in the embedding space was used directly. Key performance metrics included Macro-F1 score (for multi-label antigen classification), Recall@100 (for rare clonotype retrieval), and Area Under the Precision-Recall Curve (AUPRC, focusing on rare classes). All reported results are the mean and standard deviation across 3 independent runs with different random seeds.

## G.2 RESULTS AND DISCUSSION

The comparative performance across the selected models is summarized in Table 6.

Table 6: Performance comparison on antigen classification and rare clonotype retrieval tasks.

| Model | Antigen Macro-F1 | Rare Recall@100 | AUPRC |
|---|---|---|---|
| ESM-2-150M(Lin et al., 2023) | $0.627 \pm 0.011$ | $0.481 \pm 0.018$ | 0.601 |
| ProtST-ESM-1B(Xu et al., 2023) | $0.645 \pm 0.009$ | $0.503 \pm 0.015$ | 0.618 |
| ImmunoGraph (Ours) | $\mathbf{0.712 \pm 0.006}$ | $\mathbf{0.594 \pm 0.010}$ | $\mathbf{0.681}$ |

ImmunoGraph's representation learning module demonstrated superior performance across all metrics compared to the established baselines. This suggests that the multi-modal fusion mechanisms and the antigen-aware pre-training inherent in the ImmunoGraph pipeline facilitate the learning of more discriminative and biologically meaningful embeddings. The enhanced capability to identify rare clonotypes is particularly noteworthy, as it aligns with the framework's overarching design principle of equitable representation for minority subgroups, even before the application of explicit fairness constraints during clustering.

## H VISUALIZATION

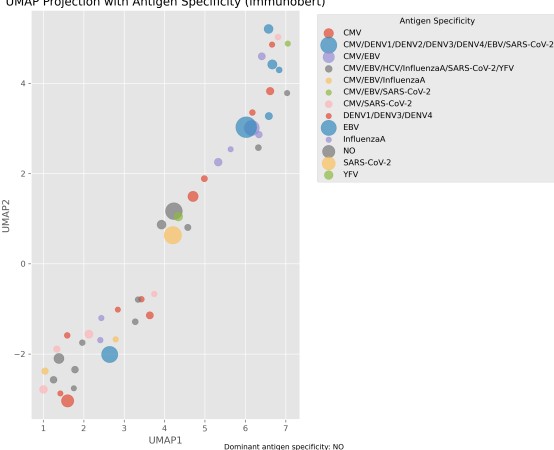

Figure 4: UMAP projection of ImmunoBERT embeddings showing conserved antigen clusters.

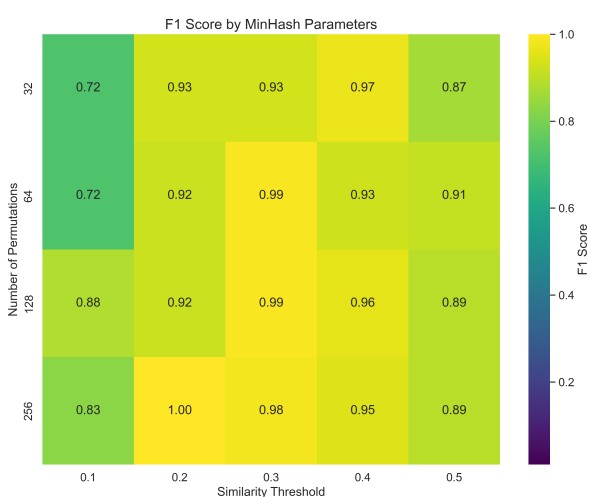

Figure 5: F1 Score Heatmap for MinHash Parameter Selection

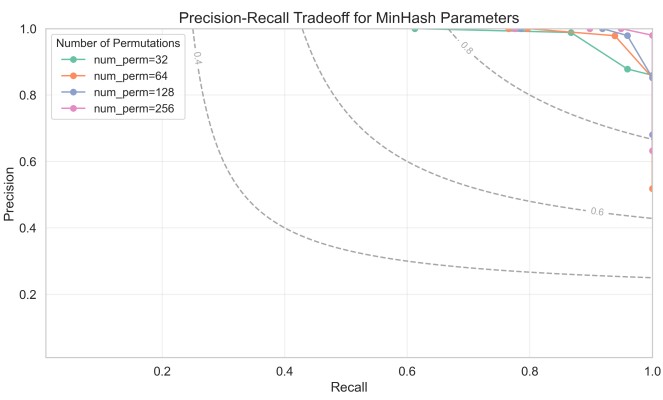

Figure 6: Parameter optimization landscape for MinHash configurations.

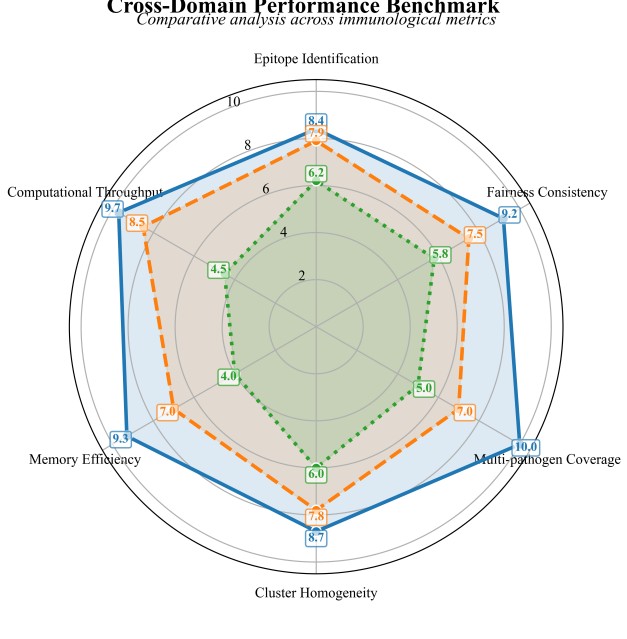

Figure 7: Performance enhancement across computational domains.

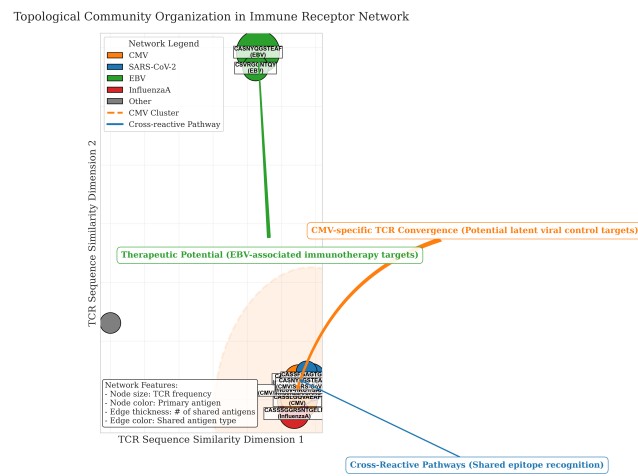

Figure 8: Topological community organization in immune receptor network. Node size indicates TCR frequency, node color indicates primary antigen, edge thickness represents the number of shared antigens, and edge color denotes shared antigen type. Key pathways are highlighted for CMV-specific TCR convergence (orange), EBV-associated immunotherapy targets (green), and cross-reactive pathways (blue).

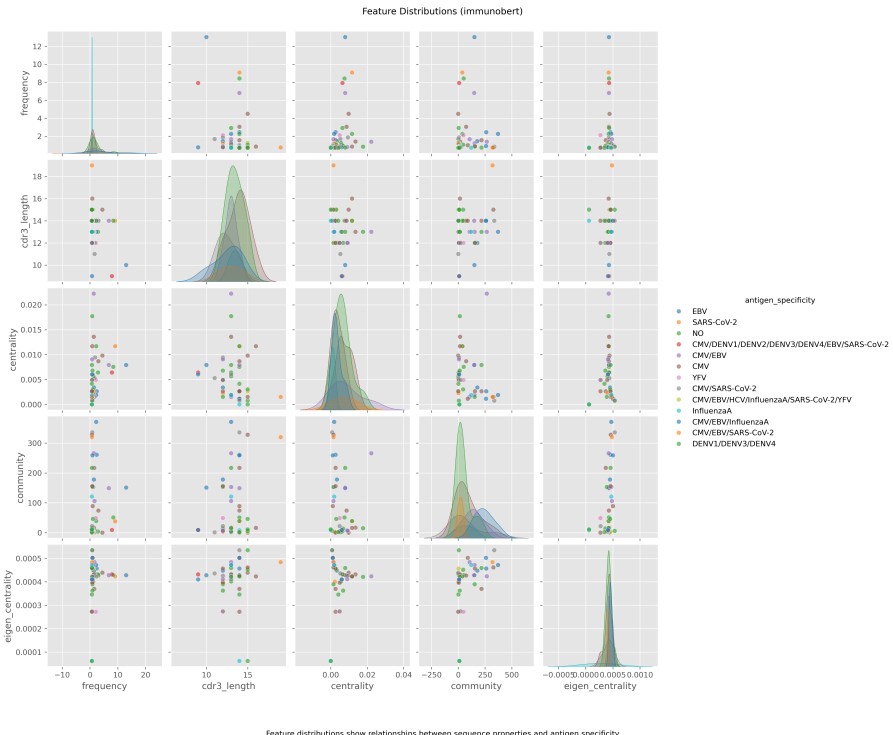

Figure 9: Feature distributions of immune receptor sequences across different antigens. Each subplot compares two features among frequency, CDR3 length, centrality, community, and eigen centrality. Colors indicate antigen specificity.

## I  EXPERIMENTAL CONFIGURATION

All datasets (VDJdb, McPAS-TCR, ImmuneCODE) are publicly available at their respective portals.

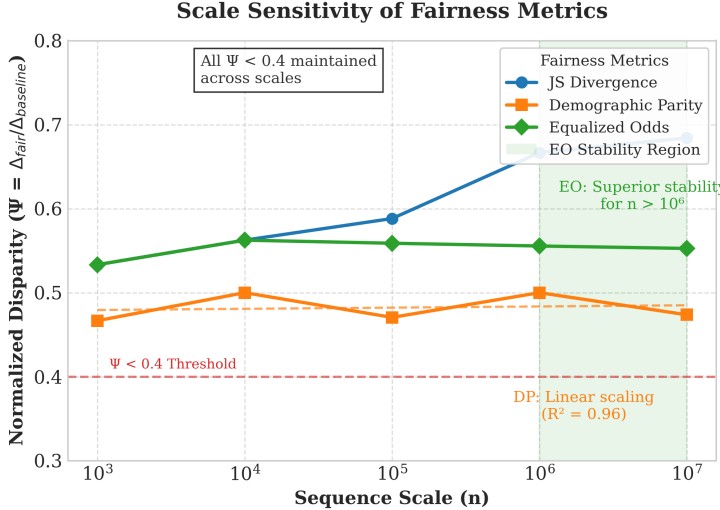

Figure 10: Scale sensitivity of fairness metrics. Normalized disparity ($\Psi = \Delta_{\text{fair}}/\Delta_{\text{baseline}}$) is plotted for JS divergence, demographic parity, and equalized odds across sequence scales. The green shaded region highlights the stability of equalized odds for large-scale settings.

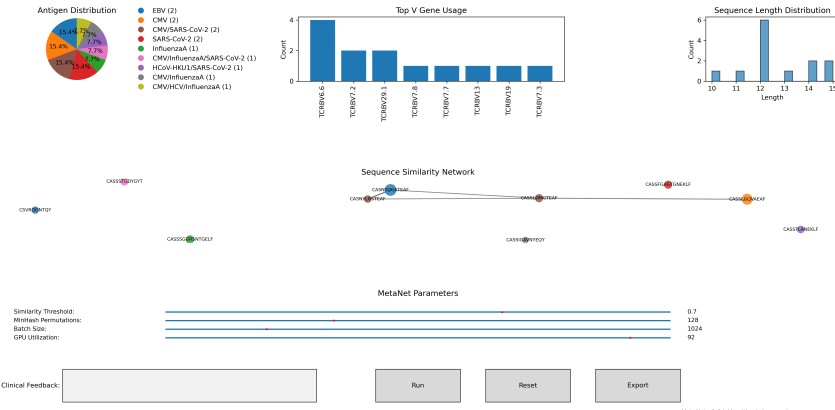

Figure 11: Clinical decision support dashboard with human-AI collaboration.

Table 7: Experimental configuration summary.

| Parameter | Configuration | Source |
|-----------|---------------|--------|
| Receptor Sequences | TCR $\beta$-chain repertoires | VDJdb(Shugay et al., 2018) |
| CDR3 Variants | 2.65K (compact), 1.2M (extended), 1M (scalability) | McPAS-TCR(Tickotsky et al., 2017) |
| Oncogenic Targets | 48.7K tumor neoantigens | ImmuneCODE(Nolan et al., 2025) |
| Cross-Domain Tooling | PyTorch-Geometric, Fairlearn | ML Commons |
| Biological Interactions | 25.1K pairwise associations | COVIDSeq |
| Computational Infrastructure | NVIDIA A100 (80GB), Dual Xeon 6348 | - |
| Evaluation Metrics | Efficiency, Memory, Accuracy, Fairness | - |

