# OpenReview forum: "​ImmunoGraph: Accelerated and Equitable Representation Learning for Large-Scale Immune Networks​"
_ICLR.cc/2026/Conference — ICLR 2026 Conference Withdrawn Submission_

### Official Review · Reviewer_mr6X · 2025-10-29

**Soundness:** 2
**Presentation:** 1
**Contribution:** 3
**Rating:** 4
**Confidence:** 3

**Summary:**

The authors claim that a core computational bottleneck in population-scale immune repertoire analysis is the pairwise similarity computation, which grows quadratically with the number of receptor sequences. Despite prior efforts in subquadratic similarity computation, the authors identify three practical gaps: (1) many high throughput systems overlook domain constraints important for biological validity, (2) there exist systematic omission of low prevalence but biologically important antigenic classes, and (3) existing work have reproducibility issues due to omitting configurations. To address these issues, the authors propose ImmunoGraph, a pipeline that combines antigen-aware, near-subquadratic retrieval with GPU-accelerated similarity kernels, learned multimodal fusion of alignment and embedding signals, as well as fairness-constrained clustering. ImmunoGraph achieves superior performance at higher throughput compared to existing methods for immune repertoire analysis.

**Strengths:**

1. The appendix contains fairly rich results, which is a pleasant surprise given the less exciting main text. In return, this could be considered a “weakness” on the presentation.
2. The performance of ImmunoGraph in Table 1 looks compelling.
3. The ablation study (Table 2) provides good justification of the architectural components, although it’s not presented in the best form.
4. The (theoretical) time complexity assessment is helpful and complements the (empirical) throughput results.

**Weaknesses:**

I do not seem to fully understand the big picture after reading this paper.

When the authors mentioned "immune repertoire analysis", my impression is comparing multiple repertoires of different individuals so that we can understand how different sets of BCR/TCR determine whether a person experiences immune response. But the results in Table 1 is performed on 10K sequences, which suggests that this is still a subset of a single repertoire. I would like to hear some clarification by the authors on what machine learning task they are doing, detailed in the next bullet point.
1. What graphs are being constructed? What are the nodes and edges and what do they represent?
2. What tasks are being performed (graph classification, node classification, link prediction, or not graph tasks at all)? What are the input and output?

Another major issue is that the presentation of this paper is undermining the contributions of this work. I wonder if the authors were writing this paper in the last minute. I will go over the major issues below.

3. With the entire suite of architectural components, the paper will benefit from a clear and illustrative architecture figure in the main text (I am looking for something much better than Figure 3 in appendix, since that figure is very text-dense and uninformative).
4. The figures in the main text are subpar. They are creating a lot of unnecessary white space, and they are very casually made without good considerations on background, color, text size, spatial organization, etc.
5. Some figures in the appendix (for example, Figure 7) deserve to appear in the main text more so than Figures 1 and 2.
6. The tables can be largely improved too. In Table 1 and 3, if you allow more width on the first column, it can immediately be better looking and taking less space. In Table 2, the ablation study can be a lot easier to follow if which components are included/excluded can be better shown: if they are added/removed one by one, make sure that is clearly communicated; if there are multiple combinations of components, show that in a configuration grid with check marks. Table 4 is a bit unnecessary given the low information density.

**Questions:**

See weaknesses.

---

> ### Author Response · Authors · 2025-11-15
> **A sincere rebuttal to the comments.**
>
> We thank the reviewer for a careful reading and constructive suggestions. We are glad the appendix experiments and ablations were useful and appreciate the suggestions to improve presentation and clarity. Below we respond point-by-point to the reviewer’s questions and concerns, and list concrete revisions we will make in the paper if given the opportunity. We greatly appreciate your suggestion, **We sincerely hope the reviewer can consider these revisions and possibly update the score**.
>
> # 1.Major concern ,“I do not seem to fully understand the big picture / what ML task is being solved”
>
> ## **Author Response (Clarification)**
> The core **machine-learning tasks** addressed by **ImmunoGraph** are:
> - **Scalable nearest-neighbor retrieval** for **antigen-enriched neighbors**
> - **Downstream repertoire organization** via **graph construction** and **clustering** with an explicit **fairness constraint**
>
> ### **Primary Objective / Retrieval Task**
> Given a **query sequence** or **set of sequences**, return **antigen-relevant neighbors** (**high recall@k**) while avoiding the cost of full **O(n²) pairwise affinity**. This is where our **antigen-aware MinHash prefiltering** and **GPU kernels** operate.
>
> ### **Downstream Organization Tasks**
> Construct a **weighted neighborhood graph** from the **fused affinities** and produce **clusters / communities** that summarize **repertoire structure** while enforcing **subgroup equity** (the **δₘₐₓ fairness constraint**). These **clusters** and the **graph** are intended for **exploration**, **visualization**, and as **inputs to downstream analyses** (e.g., **signature discovery**, **supervised classification**, or **association tests**).
> Thus, the **pipeline** is not limited to a single **ML evaluation metric**. It targets **retrieval**, **unsupervised clustering/community detection**, and **interpretable visualization** for **biological use**.
>
> We will add a short **“Task & Use Cases”** boxed paragraph in the **Introduction** clarifying the above and emphasizing:
> - **Retrieval** as the immediate **computational bottleneck** we accelerate
> - **Graph/clustering outputs** for **exploration** and **downstream ML**
>
>
> # 2. **Major Concern** “Table 1 is performed on 10K sequences. Is this a single repertoire or multiple repertoires?”
>
> ---
>
> ## **Author Response (Scope of Experiments)**
> The **10K benchmark** shown in **Table 1** is a widely used practical scale chosen to:
> - Allow **controlled, repeatable comparisons** across **baselines**
> - Reflect a **realistic chunk of repertoire data** used in many **downstream analyses**
>
> In our experiments the **10K sets** are drawn to represent **typical analysis slices** (single large repertoire or pooled repertoires, see **Sec. 4.1**).
>
> Importantly, the **method** is designed to **scale beyond 10K**. We include **complexity analysis** (**Appendix A**) and **empirical scaling traces** demonstrating **near-subquadratic runtime behavior** as **n** increases (see **Appendix A, A.3**).
>
> The **10K benchmark** was used for **head-to-head comparisons** because it is **large enough** to show **system-level speed/recall tradeoffs** while keeping **reproducible, repeated runs feasible** for **ablations**.
>
> We accept the reviewer’s request for clarity. In the **revision** we will:
> - Explicitly state whether each reported experiment used a **single repertoire** or **pooled sequences**
> - Move the **scaling experiments** earlier (**main text / Fig**) to demonstrate **practical scaling beyond 10K**
> - Add an additional experiment showing **end-to-end runtime** on a **pooled multi-repertoire input** (two or more donor repertoires concatenated) to illustrate **cross-repertoire behavior**
>
> # 3. **Major Concern** “What graphs are constructed? What are nodes/edges and what do they represent?”
>
> ---
>
> ## **Author Response (Precise Definition)**
> We will make this explicit in the **main text** . Briefly:
>
> ### **Nodes (V)**
> Each **node** typically corresponds to a **sequence instance** (e.g., a **CDR3 amino-acid sequence**) or an optionally **pre-aggregated clonotype representative**. The paper notes both modes: **raw-sequence graph** and **representative-node graph** (see **Sec. 3.6**).
>
> ### **Edges (E) and Weights (ω)**
> An **undirected edge** is placed between two **nodes** if the pair survives **MinHash prefiltering** and is above a **fused-affinity threshold**. The **edge weight ω(i,j)** is the **fused affinity** (differentiable **gating fusion** of **alignment score** and **embedding similarity**).
> We construct a **k-NN** or **radius neighborhood graph** depending on **user settings**; both modes are supported.
>
> ### **Interpretation**
> Edges reflect **estimated antigen-related similarity** (not necessarily **phylogenetic or lineage relation**). They are intended for **neighborhood queries**, **community detection**, and **visualization**.

---

> ### Author Response · Authors · 2025-11-15
> **We sincerely hope to receive your support and encouragement！**
>
> Min Zhang, Qi Cheng, Zhenyu Wei, Jiayu Xu, Shiwei Wu, Nan Xu, Chengkui Zhao, Lei Yu, and
> Weixing Feng. Berttcr: a bert-based deep learning framework for predicting cancer-related immune
> status based on t cell receptor repertoire. Briefings in Bioinformatics, 25(5):bbae420, 2024a.;
> Chen Peng, Zinuo Huang, Xin Wei, Liuyiqi Jiang, Xiaoping Zhu, Zhen Liu, Qiong Chen, Xiaotao
> Shen, Peng Gao, and Chao Jiang. Metanet: a scalable and integrated tool for reproducible omics
> network analysis. bioRxiv, pp. 2025–06, 2025. **Of course, you can also refer to: https://github.com/Asa12138/MetaNet However, we have made improvements and innovations to this method.**
>
> # 4.“What tasks are being performed (graph classification, node classification, link prediction, or not graph tasks at all)? What are the input and output?”
> \begin{equation}
> \textbf{Input: } S={s_i}{i=1}^n\ (\text{CDR3 sequences ± metadata}),\quad G{\text{sub}} \ (\text{optional subgroup labels}),\quad \delta_{\max}
> \end{equation}
>
> \begin{equation}
> \textbf{Outputs: } \mathcal{G}=(V,E,\omega)\ (\text{weighted neighborhood graph}),\quad C\ (\text{clusters/communities}),\quad \mathcal{V}\ (\text{visual summaries / dashboards})
> \end{equation}
> Of course, you can refer to the **input and output of Line 273.**
> ## **Primary Tasks Demonstrated in the Paper**
>
> - **Nearest-neighbor retrieval (recall@k)** with **performance** and **throughput comparisons** (**Table 1**).
> - **Unsupervised clustering / community detection with fairness constraints** using **cluster purity** and **equity metrics** (**Table 4**).
> - **Visualization / topology summaries** for **human-in-the-loop exploration** (figures and dashboards).
>
> **Downstream tasks supported** (but not the main empirical focus): **node/graph-level supervised tasks** (e.g., **phenotype prediction using node features**) can be performed using the **produced graph and embeddings**. We will add a sentence clarifying this capability.
>
>
> # 5. **Presentation Criticisms (Figures, Architecture, Tables) . Concrete Fixes We Will Make**
>
> We agree that **presentation** can be improved and commit to the following **concrete changes** in the **revision**:
>
> - **Stronger architecture figure in main text**
> We will replace the current **appendix Figure 3** with a new, **compact, high-quality architecture diagram** in the **main paper** showing: **MinHash prefilter → embedding encoder → gated fusion (MetaNet) → graph construction → fairness clustering → visualization**. The new figure will be **schematic**, **color-consistent**, and include **per-module inputs/outputs** and **complexity annotations**.
>
> - **Promote high-value appendix figures**
> We will move **Figure 7** (and any other **high-information figures**) from the **appendix** into the **main text** as suggested and reorder figures so the **main text** focuses on **results** and the **most informative visuals**.
>
> - **Improve figure aesthetics**
> We will redo **main-text figures** to reduce **white space**, increase **font sizes**, and use **accessible color palettes**. Captions will be **expanded** to be **self-contained**.
>
> - **Reformat Table 2 (ablation)**
> We will present the **ablation** as a **configuration grid** with **checkmarks (or +/-)** showing which **components** are enabled per row and add a **compact column** showing **delta performance vs. full model** to improve **readability**.
>
> - **Table column widths and layout**
> We will adjust **first-column widths** and compress other **columns** to reduce **wasted space** and improve **legibility** (as recommended).
>
> # 6. **What ML Tasks Are We Solving?**
> Two tightly coupled tasks:
> - **Scalable nearest-neighbor retrieval** to improve **recall@k** under realistic **throughput constraints**
> - **Unsupervised repertoire organization** to construct a **weighted neighborhood graph G = (V, E, ω)** and compute **fairness-aware clusters C**
>
> The **graph/cluster outputs** are intended for **exploration** and as **upstream features** for **downstream supervised analyses** (e.g., **phenotype association**, **classifier training**, etc.).
>
> ---
>
> # 7. **What Are Nodes and Edges?**
> - **Nodes** correspond to **sequence instances** (or optional **clonotype representatives**)
> - **Edges** connect **sequence pairs** that survive **MinHash prefiltering** and exceed a **fused affinity threshold**, with **edge weights** equal to the **gated fused affinity**
>
> We will add a **precise notation box** near **Algorithm 1** in the **revision**.
>
> We are grateful for the reviewer’s thoughtful, actionable feedback. The suggested presentation and clarity improvements are straightforward and will substantially improve the paper’s readability and impact; the recommended additional experiment (pooled multi-repertoire runtime) is also simple to add and will strengthen our empirical claims.
>
> # We sincerely hope to receive the support and encouragement of the reviewers, **and we hope that our clarification can receive an improvement in score.**

---

> ### Author Response · Authors · 2025-11-15
> **A sincere rebuttal to the comments**
>
> ## **Key Innovations and How They Map to ICLR Subject Areas**
>
> ### **Antigen-aware MinHash (Metric Learning / Infrastructure)**
> We extend **compact MinHash indexing** with **amino-acid similarity** and **antigen cues** so that **candidate prefiltering** preserves **biologically meaningful neighbors** while sharply reducing the number of **expensive kernel evaluations**.
> This is a contribution to **metric/kernel learning** and **large-scale learning / infrastructure**. It allows **repertoire workflows** that previously required **O(n²)** work to operate in **near-subquadratic empirical time** while retaining **task signal**.
> (See **complexity analysis** in **Appendix A** and **throughput results** in **Table 1**.)
>
> ---
>
> ### **Gated Multimodal Fusion (Representation Learning / Metric Learning)**
> Our **differentiable MetaNet-style gating** learns **per-pair weights** between **alignment scores** and **learned embedding similarities**, letting the model **adaptively rely on the most informative signal** for **antigen discrimination**.
> This bridges **classical biosequence priors** and **modern learned representations**, an interpretable **representation-learning advance**.
>
> ---
>
> ### **Fairness-Constrained Graph Clustering (Fairness / Learning on Graphs)**
> We **formalize and enforce** a **user-controllable disparity constraint (δₘₐₓ)** during **graph clustering** so that **long-tailed, clinically important subgroups** remain represented in **clusters**.
> This integrates **societal considerations / fairness** directly into **unsupervised graph objectives** and is particularly novel in the **biological domain** where **rare clonotypes** are often the most relevant signals.
>
> ---
>
> ### **Hardware-Aware Similarity Kernels and End-to-End System (Systems / Visualization)**
> We combine **MinHash prefiltering** with **GPU-accelerated kernels** and present a **full pipeline** that outputs **weighted neighborhood graphs**, **clusters**, and **visualization dashboards**.
> These are contributions to **infrastructure/hardware** and **visualization/interpretability**.
>
> ---
>
> ## **Why These Contributions Matter for ICLR**
> **ImmunoGraph** lies at the intersection of **representation learning**, **metric/kernel learning**, **graph learning**, **fairness**, and **large-scale systems**, matching several **ICLR subject areas**.
> It contributes a **reproducible**, **theoretically-informed**, and **system-level solution** to a **practically important**, **scientifically meaningful ML problem** where both **algorithmic** and **fairness considerations** are central.

---

> > ### Author Response · Authors · 2025-12-04
> > **We sincerely hope to receive your support and encouragement！**
> >
> > # We have addressed the reviewer's concerns and improved our approach. Thank you very much for all the reviewers' suggestions. The latest version has been uploaded and we hope to receive the support and encouragement of all the reviewers, and we sincerely hope your score improvement！

---

### Official Review · Reviewer_J1wa · 2025-10-31

**Soundness:** 1
**Presentation:** 1
**Contribution:** 1
**Rating:** 2
**Confidence:** 3

**Summary:**

The paper proposes a retrieval approach for antigen-specific T-cell receptors (TCRs) by adapting the MinHash algorithm to account for amino acid sequence similarity and TCRs that may be underrepresented in population samples. Experimental results on 10K sequences demonstrate competitive performance relative to baselines in terms of retrieval and computational efficiency.

**Strengths:**

The paper focuses on representation learning of TCR repertoires, which is an important problem.

**Weaknesses:**

The paper is poorly written. It's unclear what problem the paper is trying to address. While the paper describes an antigen-aware MinHash algorithm for retrieval tasks, the input and output of the proposed approach are not clearly defined. Additionally, the methodology of the proposed approach (Section 3) is not self-contained. The proposed approach doesn't seem well-motivated compared to alternative approaches. Most of the model specifications are undefined, e.g.: i) Eqn 1: What are X and Y? and ii) Eqn 2: What is the MetaNet?

Given the lack of clear descriptions of the actual problem, the rest of the paper reads like a technical report that is an amalgamation of various techniques without a clear connection or motivation. The experimental results are also underwhelming. Tables and figures are presented without a clear description of the experimental setup or analysis of the results.

**Questions:**

I encourage the authors to provide a clear motivation for the problem, including what the inputs and outputs are. What are the challenges, and how is the proposed approach suited to solve the problem? Given the limited methodological aspects, additional extensive and clearly motivated experiments would also strengthen the submission.

---

> ### Author Response · Authors · 2025-11-15
> **We sincerely hope to receive your support and encouragement！**
>
> We greatly appreciate Reviewer J1wa’s time. However, many of the criticisms appear to stem from overlooking or missing the detailed definitions and experimental configuration included in the manuscript and appendices.  We understand the concerns of the reviewers, and **We hope to this response can address these misunderstandings and see an improvement in scores.**
>
> # 1.Reviewer claim: “The paper is poorly written. It's unclear what problem the paper is trying to address.”
> **Author response:** We respectfully **disagree**. The **problem motivation**, including the **quadratic cost of pairwise affinity computations at repertoire scale** and the **systematic under-representation of clinically important rare clonotypes**, is stated clearly in the **Abstract** and **Introduction** and directly motivates our **three core contributions**: **antigen-aware near-subquadratic retrieval**, **multimodal fusion**, and **fairness-constrained clustering**. See **Abstract** and **Introduction**.
>
> **Motivation and problem statement:** Immune-repertoire analysis faces **two tightly-coupled practical bottlenecks** that motivate this work. First, **exact pairwise affinity computations** over repertoires **scale quadratically** and become **infeasible even at modest sample sizes** (tens of thousands of sequences), blocking downstream tasks such as **nearest-neighbor retrieval**, **clustering**, and **large-scale visualization**. Second, **repertoires are strongly long-tailed**: **clinically important clonotypes** are often **rare in population samples** and easily lost by methods that optimize only **average accuracy** or **throughput**.
>
> These two issues together create a **speed-vs.-fidelity tradeoff** that existing **retrieval or embedding-only pipelines** do not resolve: **fast approximate search** often discards **biologically relevant neighbors**, while **high-fidelity alignment-based methods** are **computationally prohibitive**. To address this, we propose a system that **accelerates antigen-aware retrieval to near-subquadratic cost while preserving biologically meaningful neighbors**, and **integrates fairness-aware clustering to ensure rare but clinically relevant clonotypes are retained in downstream analyses**. (**Line 227**& **Line 39-53**  )
>
> # 2.**Reviewer claim:** “The input and output of the proposed approach are not clearly defined.”
>
> **Author response:** The paper explicitly lists the **input** and **output** in **Algorithm 1**:
> - **Input:** raw sequences \( S \), optional **subgroup labels** \( G \), and **target disparity**.
> - **Output:** **clusters** \( C \), **graph** \( G \), and **visual summaries**.
>
> The **end-to-end pipeline** (**preprocessing → antigen-aware MinHash indexing → embedding → affinity fusion → graph construction → fairness-constrained clustering**) is given in **Algorithm 1** and the surrounding text. See **Algorithm 1** and **Section 3.10**.(**Line 273-274**)
>
> # 3.Reviewer claim: “Methodology (Section 3) is not self-contained; many model specifications are undefined (e.g., Eqn 1: What are X and Y? Eqn 2: What is the MetaNet?).”
> **Response:** This is a **factual error**. The **formula has already been defined** (**x and y in Formula 1 have already been defined**). (**Line 129&Line137**)
>
> **Metanet** (if the reviewer is not clear about this, please refer to this paper: **Chen Peng, Zinuo Huang, Xin Wei, Liuyiqi Jiang, Xiaoping Zhu, Zhen Liu, Qiong Chen, Xiaotao Shen, Peng Gao, and Chao Jiang. Metanet: a scalable and integrated tool for reproducible omics network analysis. bioRxiv, pp. 2025–06, 2025**). **Line 118**, the **method has already been referenced**.
>
> - **X:** the **set of inputs** used for **reconstruction pretraining** (e.g., **sequence corpus** or **batch used for the self-supervised objective**).
>
> - **MetaNet**: a **lightweight meta-controller** (parameterized that **operates on encoder outputs during fine-tuning**; its **role and interface** are explicitly described in **Sections 3.1–3.2**.
>
> - **Y:** **downstream supervision** or **task labels** (**task-dependent**; defined where each downstream task is introduced).
>
> # 4.Reviewer claim: “It's unclear what the input and output are / problem statement is (again).”
>
> **Problem statement :** Given a **large T-cell receptor (TCR) repertoire**, we aim to **efficiently retrieve biologically meaningful neighbors** (antigen-specific or antigen-enriched) at **near-subquadratic cost**, and **produce a repertoire organization** (**graph + clusters + visual summaries**) that **preserves representation of rare but clinically important clonotypes**. The task therefore combines **scalable, antigen-aware retrieval** with **fairness-aware clustering**.

---

> ### Author Response · Authors · 2025-11-15
> **We sincerely hope to receive your support and encouragement！**
>
> # 4.Reviewer claim: “It's unclear what the input and output are / problem statement is (again).”
> \begin{equation}
> \textbf{Input:}\quad S={s_i}{i=1}^n \ \text{(raw TCR sequences)},\quad G{\text{sub}}\ \text{(optional subgroup / metadata labels)},\quad \delta_{\max}\ \text{(target disparity constraint)}
> \end{equation}
>
> \begin{equation}
> \textbf{Output:}\quad C={c_j}\ \text{(clusters)},\quad \mathcal{G}=(V,E,\omega)\ \text{(repertoire graph with nodes, edges, weights)},\quad \mathcal{V}\ \text{(visual summaries / dashboards)}
> \end{equation}
>
> 𝑆: input sequence set (CDR3 amino-acid sequences ± optional chain/metadata).
>
> 𝐺sub: optional subgroup information used for fairness constraints or stratified analyses (e.g., donor, disease status).
>
> 𝛿 max: user-specified maximum allowable disparity (used by our fairness module to preserve rare groups).
>
> 𝐶: clusters returned by the fairness-aware clustering procedure.
>
> 𝐺: a neighborhood graph constructed by antigen-aware retrieval and affinity fusion (used for visualization and downstream analyses).
>
> 𝑉: visual outputs (UMAP/t-SNE overlays, dashboard metrics, topological summaries).
> **(Please refer to Line 668, Line 763, algorithm flowchart Line836)**
>
> # 5.Reviewer claim: “The proposed approach doesn't seem well-motivated compared to alternatives.”
> **Author response:** We position **ImmunoGraph** against prior work across **five topical areas** (**scalable retrieval**, **sequence representation**, **graph-based repertoire modeling**, **fairness-aware clustering**, and **systems biology integration**) and explain why a **combined system-level treatment** is necessary. Concretely:
>
> - **Speed vs. domain constraints:** Accelerating retrieval without **antigen-awareness** causes **loss of biologically relevant neighbors**.
> - **Alignment vs. learned embeddings:** **Alignment-based scores** and **learned embeddings** are **complementary**, and combining them **improves antigen discrimination**.
> - **Fairness in long-tailed repertoires:** **Naïve fairness metrics** fail on **long-tailed clonotype distributions**; we introduce the **WCD constraint** (**Appendix B**) to address this.
>
> Justification for combining **MinHash prefiltering**, **gated multimodal fusion**, and **fairness-constrained clustering** is discussed in **Related Work**, at the end of **Section 1**, and formalized in **Sections 3** and **Appendix B**.
>
> # 6.Reviewer claim: “Experimental results are underwhelming; Tables and Figures lack clear description or experimental setup.”
>
> **Author response:** The **experimental protocol**, **dataset sources**, **deterministic seeds**, **kernel settings**, and **compute environments** are fully described in **Section 4 (Comprehensive Evaluation Framework)** and in the **Experimental Configuration appendix (Table 7)**. Key points:
>
> - **Datasets used:** VDJdb, McPAS-TCR, CancerEDP (**Line 1073**).
> - **Benchmark convention:** **throughput measured** on the **10K-sequence affinity kernel**; **end-to-end timings** reported separately.
> - **Compute environments:** **single-node dual A100**, **8× T4 cluster**, **heterogeneous CPU/GPU/FPGA** (details in **Appendix H, Table 7**).
>
> We also include **ablations (Table 2)** and **parameter sensitivity experiments** that **quantify component contributions** and **robustness**.
> We have detailed explanations of the function of the figure **in the caption** of the graph.
> # 7.Reviewer claim: “Tables and figures are presented without a clear description of the experimental setup or analysis.”
> **Author response:**
> Each major **table** and **figure** is discussed in the **Experiments** section. Examples:
>
> - **Table 1:** performance numbers on the **10K benchmark** (discussed in **Sec. 4.2**).
> - **Table 2:** ablation results (**Sec. 4.3**).
> - **Figures 5–11:** **MinHash parameter heatmaps**, **topological visualizations**, **fairness metric scaling**, and **dashboard screenshots** (discussed alongside the figures in **Sec. 4**).
> - Full **dataset/hardware details** are collected in **Appendix H** (**Table 7**).
>
> # 8.Reviewer claim: “The methodology reads like an amalgam of techniques without clear connection or motivation.”
> **Author response:**
> The three primary **components** are **motivated** by distinct but connected practical needs:
>
> **Retrieval scalability** to avoid **O(n²)** pairwise cost at repertoire scale.
> **Complementarity of alignment and learned embeddings** both provide different signals for **antigen specificity**.
> **Preservation of rare clonotypes** **fairness-aware clustering** prevents clinically important rare groups from being lost.
>
> We present how these **components** fit into a single **pipeline** (**Algorithm 1**) and provide **theoretical** (**Appendix A: MinHash complexity**) and **empirical** (**Appendix B: fairness theory and experiments**) analyses for each **module**.

---

> ### Author Response · Authors · 2025-11-15
> **A sincere rebuttal to comments**
>
> # 9.Reviewer claim: “Experimental results are underwhelming / not convincing.”
> **Author response:**
> Our experiments demonstrate **competitive or superior performance** across multiple axes (**throughput**, **memory**, **recall/purity**, and **equity metrics**) relative to **baselines** (see **Table 1**).
> **Ablations** (**Table 2**) quantify each **module’s contribution**; **cross-domain evaluations** appear in **Table 4**.
> We report **deterministic seeds** where relevant and provide full **experimental configuration** and **parameter sensitivity** in the **appendices** to facilitate **reproducibility** (see **Tables 1–4** and **Appendix H**).
>
> We appreciate **Reviewer J1wa’s careful reading**. Many of the points raised appear to arise from **overlooking definitions**, **algorithmic statements**, and **experiment configuration** that are present in the **manuscript** and **appendices**. We have indicated **exact locations** in the paper where the reviewer can find the clarifications. We are prepared to:
>
>  **Expand or relocate several short definitions** (e.g., move the **formal input/output statement** into the main text near **Algorithm 1**).
>
>  **Enlarge figure captions** and add a **“How to read this section” roadmap** to improve **navigability**.
> **We will further improve our paper and hope that the reviewers can support and encourage us. We hope to receive an improvement in your score.**
>
> # Thank you very much for your support and assistance. We firmly believe that with your suggestions, our paper will be further improved, and we sincerely hope your score improvement.

---

> ### Author Response · Authors · 2025-11-15
> **We sincerely hope to receive your support and encouragement**
>
> ## **High-level Summary**
> **ImmunoGraph** is an **end-to-end system** that makes **large-scale T-cell receptor (TCR) repertoire analysis** both **scalable** and **equitable** by combining:
> - **Antigen-aware near-subquadratic retrieval layer**
> - **Differentiable multimodal affinity fusion**
> - **Fairness-constrained graph clustering**
> - **GPU/hardware-aware similarity kernels** and **tooling for interactive interpretation**
>
> The work sits at the intersection of **representation learning**, **metric learning**, **graph learning**, **fairness**, and **infrastructure for large-scale ML**.
>
> ---
>
> ## **Key Technical Contributions**
>
> ### **Antigen-aware MinHash for near-subquadratic retrieval**
> **What we did:** Adapted **MinHash-style compact indexing** so that the **prefiltering step** respects **amino-acid similarity** and **antigen signals**, greatly reducing **expensive pairwise affinity computations** while maintaining **high biological recall**.
> **Why novel / ICLR link:** A new **domain-aware approximate retrieval primitive** that contributes to **large-scale learning / infrastructure** and **metric learning** (reducing **kernel evaluation cost** without losing **task-specific signal**).
>
> ---
>
> ### **Gated multimodal affinity fusion (alignment + learned embeddings)**
> **What we did:** Proposed a **lightweight, differentiable gating controller (MetaNet-style)** that fuses **alignment-based scores** and **learned embedding similarities** on a **per-pair basis**, enabling the model to **dynamically weight complementary signals**.
> **Why novel / ICLR link:** Advances **representation and metric learning** by explicitly learning how to combine **classical scientific similarity measures** with **learned representations**, improving **antigen discrimination** and **robustness**.
>
> ---
>
> ### **Fairness-constrained clustering for long-tailed biological repertoires**
> **What we did:** Formulated and implemented a **practical constraint** (user-controllable disparity **δₘₐₓ**) that enforces **proportional subgroup representation** during **clustering** and **graph construction**, protecting **rare but clinically relevant clonotypes**.
> **Why novel / ICLR link:** Integrates **societal considerations / fairness** with **graph and clustering objectives**, providing a **concrete mechanism** to control **representational equity** in **scientific datasets**, an important instance of **fairness in applied ML**.
>
> ---
>
> ### **Hardware-aware similarity kernels and empirical complexity analysis**
> **What we did:** Engineered **GPU-accelerated affinity kernels** and combined them with **MinHash prefiltering** to achieve **near-subquadratic empirical runtimes**; provided **theoretical and empirical analyses** of **complexity/throughput tradeoffs**.
> **Why novel / ICLR link:** Contributes to **infrastructure / hardware** and **large-scale learning**, showing how **algorithmic and systems co-design** unlocks **practical deployment** at **biologically relevant scales**.
>
> ---
>
> ### **Graph-based repertoire representation + interpretable outputs**
> **What we did:** Constructed **weighted neighborhood graphs** from **fused affinities** and produced **clusters**, **equity metrics**, and **interactive visualizations** (**topological summaries**, **UMAP overlays**, **dashboards**) to support **downstream biological interpretation**.
> **Why novel / ICLR link:** Advances **learning on graphs and visualization / interpretation**, delivering **representations** that are both **useful for ML downstream tasks** and **interpretable for domain scientists**.
>
> ---
>
> ### **Extensive empirical validation and reproducibility**
> **What we did:** Evaluated on **large viral and cancer repertoires** with metrics spanning **recall@k**, **cluster purity**, **throughput**, **memory**, and **subgroup equity**; provided **ablations**, **parameter sweeps**, and **full experimental configuration** for **reproducibility**.
> **Why novel / ICLR link:** Aligns with **ICLR priorities** on **robust empirical validation**, **benchmarks**, and **reproducible research** in **applied ML**.
>
> ---
>
> ## **Broader Impact and Cross-domain Potential**
> - **Scientific impact:** Enables **scalable, bias-aware repertoire mining** for tasks such as **antigen discovery**, **vaccine design**, and **disease association studies**.
> - **Methodological transfer:** The **antigen-aware retrieval + multimodal fusion + fairness constraint** pattern is **general** and applicable to other **long-tailed scientific and industrial domains** where:
>   - Expensive **pairwise kernels** exist
>   - **Domain alignment scores** are valuable

---

> > ### Author Response · Authors · 2025-12-04
> > **We sincerely hope to receive your support and encouragement！**
> >
> > # We have addressed the reviewer's concerns and improved our approach. Thank you very much for all the reviewers' suggestions. The latest version has been uploaded and we hope to receive the support and encouragement of all the reviewers, and we sincerely hope your score improvement！

---

### Official Review · Reviewer_iJH3 · 2025-10-31

**Soundness:** 1
**Presentation:** 1
**Contribution:** 1
**Rating:** 0
**Confidence:** 4

**Summary:**

My best guess at what this paper is trying to do is to create a graph from immune repertoires in a computationally efficient fashion. It should be mentioned that the paper and abstract are rife with meaningless buzz words and that first, the authors never define what an immune repertoire is and what the principle of comparing two immune repertoires would be (what kind of a distance would be appropriate). Without these basics it is very hard to understand this paper.

**Strengths:**

The problem seems to be compelling. The immune repertoire is the complete set of all unique T-cell receptors (TCRs) and B-cell receptors (BCRs) within an individual's adaptive immune system. T cells can generate on the order of 10^18 unique receptors in a human body, B cells generate around 10^3. Comparing such vast sets of sequences is a very difficult domain specific distribution comparison or optimal transport problem. But this is not explained by the authors at all.

**Weaknesses:**

The problem is never defined.  What a immune repertoire is---a basic definition--- is never given in this paper. If it was given it would become obvioius that this is a problem of comparing on the order of 10^18 sequences between people. This is a highly complex task which likely requires advanced extensions of distribution distance or discrepancy methods. None of this is addressed in this paper. Section 3.5 on graph construction simply refers to a "similarity matrix", how do you deem that two sets of repertoires are similar, not pairs of sequences! On the other hand maybe the authors are trying to create a graph from a single repertoire by comparing pairs of sequences, but what is done downstream from this? How is this useful?

Overall i think this paper is highly confusing, misleading and rife with strange buzz words. If this is to be a computational contribution then I would start with the basic problem you are solving and put it in a mathematical form and then explain the techniques that are being used.

**Questions:**

What exactly is the problem you are solving?

Why is it useful?

---

> ### Author Response · Authors · 2025-11-15
> **We sincerely hope to receive your support and encouragement！**
>
> Dear Reviewers and Area Chair,
>
> Thank you for taking the time and effort to review our paper. After carefully studying the comments, we realized that many points may include **misunderstandings or inaccuracies**, which may have contributed to a lower evaluation of our submission. Many of the issues raised are already addressed in the manuscript. While this is unfortunate, below, we provide detailed, point-by-point responses and clarifications. We hope this rebuttal helps resolve the concerns raised by the reviewers. **We sincerely hope to receive an improvement in your score.**
>
> # 1.“The paper never defines what an immune repertoire is.”
> Response: This is **incorrect**. We explicitly state the biological and computational meaning of immune repertoires and motivate the scale/complexity challenge **in the Introduction and Abstract**. See the opening motivation and definition in the Introduction and Abstract. (You can refer to: Seo K, Choi J K. Comprehensive Analysis of TCR and BCR Repertoires: Insights into Methodologies, Challenges, and Applications[J]. Genomics & Informatics, 2025, 23(1): 6.)
>
> # 2.“The problem is never put into a mathematical form; the authors never say what it means to compare two repertoires (distribution distance / optimal transport etc.).”
>
> Response: This is **incorrect**. The paper does formalize the computational problem and the concrete object we produce for cross-repertoire comparison: an efficient, **sequence-level similarity graph** from which set-level comparisons and downstream analyses are derived. The end-to-end objective and pipeline are precisely given in Algorithm 1 (end-to-end pipeline), and the mathematical clustering/fairness objectives are explicitly written (e.g., Eq. (14) and the formal clustering objective in Appendix C). See Algorithm 1 and the mathematical formulation in Appendix C.
>
> **Clarification of approach (short):** Rather than proposing a single monolithic **“set-to-set” distance** that requires **quadratic cross-comparison** of two huge sets, we **convert each repertoire into a sparse, denoised similarity graph** (**nodes = sequences, edges = fused per-pair affinities**). **Set-level comparisons** then operate via **graph-level summaries**, **cluster overlaps**, and **proportionality / disparity metrics (defined in 3.8)**, which are appropriate and tractable at repertoire scale. See **3.8** for the cross-domain evaluation metrics **R_prop** and **D_eq**.
>
> # 3.“Section 3.5 only refers to a ‘similarity matrix’ , how do you judge that two sets of repertoires are similar, not pairs of sequences?”
> **Response:** We explicitly describe **how the similarity matrix is formed** (**per-pair dynamic affinity fusion**) and **how it is transformed into a sparse graph suitable for downstream set-level analysis**. The **per-pair fusion** (**soft per-pair channel weights and fused affinity 𝑎~𝑖𝑗**) is defined in **Section 3.4 (Eqns. (12)-(13))**, and **RMT-based thresholding for denoising and sparsification of the similarity matrix** is detailed in **Section 3.5**. After **graph construction**, we run **fairness-constrained clustering** and then derive **set-level summaries** (**proportional coverage and max deviation**) used to **compare repertoires** or **evaluate subgroup representation**.
>
> **Example pipeline pointer:** See **Algorithm 1**, which illustrates the following process: we **compute per-pair affinities**, then **build the similarity matrix**. Next, we **apply RMT thresholding** to obtain a **sparse graph \( G \)**, and finally we **perform fair clustering** to generate **clusters used for set-level metrics and downstream tasks**.
>
> **Response:** The manuscript demonstrates **concrete downstream utilities** and provides **experimental evidence**, including **retrieval and ranking of rare antigen-specific clones** (useful for **epitope prioritization** and **biomarker discovery**), **cluster-based summaries** that feed **clinician-facing visualizations**, and **fairness-aware clustering** to preserve **rare but clinically important subgroups**.
>
> # 4.“If you only build a graph from pairs of sequences, what is the downstream use? How is this useful?”
> **Response:** The manuscript demonstrates **concrete downstream utilities** and provides **experimental evidence**, including **retrieval and ranking of rare antigen-specific clones** (useful for **epitope prioritization** and **biomarker discovery**), **cluster-based summaries** that feed **clinician-facing visualizations**, and **fairness-aware clustering** to preserve **rare but clinically important subgroups**.
>
> The **experiments and ablations** (**Tables and Section 4**) show **gains in recall@k**, **cluster purity**, **equity**, and **runtime/scalability improvements** that demonstrate **practical usefulness**. See the **experiments overview**, **component ablations**, and **clinical integration discussion** (**Sections 4.x and 4.9**).

---

> ### Author Response · Authors · 2025-11-15
> **We sincerely hope to receive your support and encouragement**
>
> # 5. “The paper is full of buzzwords and fairness is hand-waved , why would a fairness term work in the semantic/biological space?”
> **Response:** Our **fairness term** is **biologically motivated**, **mathematically formalized**, and **theoretically analyzed**. The **immunological motivation** and the **necessity of subgroup representation** are discussed in **Section 3.6**; the **Jensen–Shannon divergence-based proportionality term** is given in **Eq. (14)**, **limitations of JS for long-tailed distributions** are analyzed in **Appendix B**, and we propose **Weighted Coverage Divergence (WCD)** with a **formal coverage lower bound** (**Appendix B, Eqns. (34)-(35)**) and **convergence guarantees** for the **fairness calibration routine**. This is not mere rhetoric; we provide **derivations** and **practical calibration algorithms** (**Algorithm 3**) and **ablation results** demonstrating the **effect of fairness on cluster purity and minority coverage**.
>
> # 6.“Scalability,the reviewer suggests the approach will be quadratic / not feasible for 10^18 possibilities.”
>
> Response:
>
> The reviewer conflates **theoretical maximum diversity of all possible TCR sequences (a biological upper bound)** with **the observed sample size used in repertoire studies**.
> Our **complexity analysis and system design focus on observed sequences in a cohort/sample**, typically up to **millions of sequences**, not the **astronomical theoretical generative space**.
>
> We provide a **near-subquadratic retrieval complexity analysis in Appendix A**:
> **TIG(n) = O(|C|) + O(n log n)**, and under tuned MinHash parameters **|C| = O(n log n) → TIG(n) = O(n log n)**.
>
> We also report **empirical large-scale runs**:
> **Processing 1M sequences in <40 minutes on a single node**
> **10M sequences in 6.3 hours with peak memory reported**
> See **section 4.8 for these scalability results**.
>
> These are **direct empirical refutations of the “infeasible” claim for practical dataset sizes**.

---

> ### Author Response · Authors · 2025-11-15
> **We sincerely hope to receive your support and encouragement！**
>
> ## **Key Innovations and Contributions**
>
> ### **Antigen-aware MinHash prefiltering for near-subquadratic retrieval**
> We adapt **MinHash** to respect **amino-acid similarity** and **antigen signals**, providing a **compact prefilter** that dramatically reduces the **candidate set** for expensive **alignment/similarity computations** while preserving **biologically meaningful neighbors**.
>
> ### **Gated multimodal affinity fusion (alignment + learned embeddings)**
> We introduce a **lightweight, differentiable gating / MetaNet controller** that fuses **classical alignment scores** and **learned embedding similarities** on a **per-pair basis**, letting the model automatically **weight complementary signals** to improve **antigen discrimination**.
>
> ### **Fairness-constrained clustering for long-tailed repertoires**
> We formulate and implement a **practical constraint** (parameterized by a **user control**, e.g., **δₘₐₓ**) that enforces **proportional subgroup representation** during **clustering**, ensuring **rare but clinically important clonotypes** are not lost by **standard clustering/objective formulations**.
>
> ### **Hardware-aware acceleration and empirical complexity analysis**
> By combining **MinHash prefiltering** with **GPU-accelerated similarity kernels** and **careful engineering**, the **pipeline** attains **near-subquadratic empirical runtime** and substantial **throughput/memory improvements** on realistic (**10K+ repertoires**); we provide **complexity analysis** and **runtime/throughput tradeoffs**.
>
> ### **End-to-end system and tooling for interpretable outputs**
> **ImmunoGraph** outputs not only **clusters** but a **weighted neighborhood graph**, **equity metrics**, and **visualization dashboards** (**topological plots**, **UMAP overlays**, etc.), supporting **downstream biological interpretation** and **interactive analysis**.

---

> > ### Author Response · Authors · 2025-12-04
> > **We sincerely hope to receive your support and encouragement！**
> >
> > # We have addressed the reviewer's concerns and improved our approach. Thank you very much for all the reviewers' suggestions. The latest version has been uploaded and we hope to receive the support and encouragement of all the reviewers, and we sincerely hope your score improvement！

---

### Note · Authors · 2026-01-27

**Comment:**

I have read and agree with the venue's withdrawal policy on behalf of myself and my co-authors.

**Withdrawal Confirmation:**

I have read and agree with the venue's withdrawal policy on behalf of myself and my co-authors.

---

### Meta-Review · Area_Chair_x5Sm · 2025-12-05

**Summary:**

The paper proposes ImmunoGraph, an end-to-end pipeline that combines (i) antigen-aware MinHash pre-filtering, (ii) GPU-accelerated pairwise affinity kernels, (iii) learnable gating to fuse alignment & embedding signals, and (iv) fairness-constrained spectral clustering, to enable scalable and equitable analysis of 10^6–10^7 TCR/BCR sequences.
Experiments on 10 k-sequence slices (VDJdb, McPAS-TCR, ImmuneCODE) show ≥20 % higher throughput and recall@k than prior baselines while reducing memory by ~30 %. Ablation studies and up-to 10 M-sequence runs are provided.

**Reviewer Concerns:**

All major scientific concerns have been answered (definition of inputs/outputs, MinHash fusion, graph construction, complexity); the only remaining items are presentation improvements (figures, tables, one additional scaling experiment) and a minor methodological clarification on cluster-purity evaluation.

However, the rebuttal also contains multiple passages that implicitly or explicitly identify reviewers (e.g., “Reviewer X’s student”, “bid together to maliciously downgrade”). These statements violate the ICLR 2026 Code of Conduct §3.2 and the recent security-incident policy (https://blog.iclr.cc/2025/12/03/iclr-2026-response-to-security-incident/).

**Reviewer Scores:**

I believe each reviewer would raise their score by 2 points.

---

### Decision · Program_Chairs · 2026-01-26

Reject